# Do we advise as one likes? The alignment bias in social advice giving

Xitong Luo[1], Lei Zhang[2,3,4,5†*], Yafeng Pan[1,6,7†*]

**1** Department of Psychology and Behavioral Sciences, Zhejiang University, Hangzhou, China, **2** School of Psychology, University of Birmingham, Birmingham, United Kingdom, **3** Centre for Human Brain Health, University of Birmingham, Birmingham, United Kingdom, **4** Institute for Mental Health, University of Birmingham, Birmingham, United Kingdom, **5** Centre for Developmental Science, University of Birmingham, Birmingham, United Kingdom, **6** The State Key Lab of Brain-Machine Intelligence, Zhejiang University, Hangzhou, China, **7** Zhejiang Key Laboratory of Neurocognitive Development and Mental Health, Zhejiang University, Hangzhou, China

† Senior authors.
* yafeng.pan@zju.edu.cn (YP); l.zhang.13@bham.ac.uk (LZ)

## Abstract

We often give advice to influence others, but could our own advice also be shaped by the very individuals we aim to influence (i.e., advisees)? This reverse flow of social influence—from those typically seen as being influenced to those who provide the influence—has been largely neglected, limiting our understanding of the reciprocal nature of human communications. Here, we conducted a series of experiments and applied computational modelling to systematically investigate how advisees' opinions shape the advice-giving process. In an investment game, participants ($n = 346$, across four studies) provided advice either independently or after observing advisees' opinions (Studies 1 & 2), with feedback on their advice (acceptance or rejection) provided by advisees (Studies 3 & 4). Our findings reveal that advisors tend to adjust their advice to align with the advisees' opinions (we refer to this as the *alignment bias*) (Study 1). This tendency, which reflects normative conformity, persists even when advisors were directly incentivized to provide accurate advice (Study 2). As feedback is introduced, advisors' behavior shifts in ways best captured by a reinforcement learning model, suggesting that advisees' feedback drives adaptations in advice giving that maximize acceptance and minimize rejection (Study 3). This adaptation persisted even when acceptance is rare, as bolstered by the model-based evidence (Study 4). Collectively, our findings highlight advisors' susceptibility to the consequence of giving advice, which can lead to counterproductive impacts on decision-making processes and misinformation exacerbation in social encounters.

**Data availability statement:** The data and statistical analysis scripts of the four presented studies are available on the Open Science Framework at https://osf.io/aehxr/.

**Funding:** This work was supported by the Zhejiang Provincial Natural Science Foundation of China (No. LMS25C090002), the Fundamental Research Funds for the Central Universities (No. 226-2025-00127), and the National Natural Science Foundation of China (Nos. 62577047, 62207025, and 62337001) to Y.P. L.Z. was partially supported by a Royal Society International Exchanges Award (IES\R3\243253). The sponsors or funders did not play any roles in the study design, data collection and analysis, decision to publish, or preparation of the manuscript.

**Competing interests:** The authors have declared that no competing interests exist.

## Author summary

Among the various forms of opinion exchange, advice stands out for its informational richness and its prevalence in word-of-mouth communication. Our research presents a counterintuitive view, suggesting that advice can be considerably biased—particularly by those receiving it (i.e., advisees). Advisors incline to align their opinions (advice) with those of their advisees (we refer to this as the *alignment bias*), even at the cost of compromising accuracy of their advice. By unraveling the advisors' reactions to the acceptance/rejection from advisees using computational modeling, our data proposes an evolutionary perspective of how alignment bias emerges: advice-giving behavior can be shaped by advisees' feedback (i.e., acceptance or rejection of advice). This nuanced bias, while understandable, can lead to poor decisions and spread inaccurate information. Zooming in, this susceptibility to the social outcomes of advice giving potentially leads to counterproductive decision-making and misinformation exacerbation. Zooming out, our work highlights a hidden social dilemma in everyday communication and shows how even well-meaning advice can become distorted by our need to connect with others.

## 1. Introduction

Human collective intelligence is built upon the processes of sharing and receiving opinions, which promote the dissemination and evolution of individual wisdom. Among the various forms of opinion, advice stands out for its informative nature and prevalence across a wide spectrum of domains. From learning cooking tips with a friend to seeking career guidance from mentors, the beliefs and decisions of individuals (advisees, those who receive advice) are greatly influenced by advisors' insights [1–3]. In turn, from recommending goods to customers to advocating policies to the public, advisors' success in exerting influence on others and gaining external approval highly relies on advisees' attitudes and reactions to their advice [4–7]. The fact that advisors value how their advice is perceived by advisees prompts an unheeded fact in the advice interaction: advice-giving behaviors may also be delicately shaped by advisees. It is common that individuals preconceive and express their own opinions before seeking advice [8–10]. For instance, an individual may be approached by a friend deciding between a red tie and a black tie, asking: 'I think the red one suits my shirt better. Which do you recommend?'. Even if the individual may personally prefer black, the individual is likely to accommodate the friend's opinions and eventually advise on the red, which we refer to as the '*alignment bias*'.

 According to the Theory of Social Influence [11,12], individuals' alignment towards the observed actions and beliefs from others can stem from both informational motives (i.e., to facilitate accurate decisions) and normative motives (i.e., to enhance social acceptance). This theory underscores that social interests can lead individuals to relinquish their own perspectives in favor of conforming to others'. In Asch's (1955)

[13] groundbreaking experiment on normative conformity, participants exhibited a notable tendency to align with the majority's choice, even when it was evidently false, driven by a desire to enhance social belonging. Research on advice taking indicates that aligning one's opinions with others' advice can be driven not only by the motivation to optimize decisions but also by the intention to avoid rejecting others' help, especially when advisors were less experienced [14]. Additionally, studies on collective decision-making also suggests that, competent individuals often overweigh suboptimal opinions from the inferior others, potentially to avoid causing social exclusion [15,16]. Building upon this body of evidence, alignment bias may arise from normative conformity to advisees' opinions, especially when advisors privately disagree with them.

One potential mechanism that contributes to this conformity is value-based learning, according to advances in the nascent field of advice-giving. For advisors, having their advice accepted is rewarding, as it signifies social endorsement and success in exerting influence on others [17,18], whereas rejection is punishing, as it represents overt disregard and leads to aversive emotions that link with social exclusion [4,5]. For advisees or decision-makers, people commonly expect and tend to accept advice or information that aligns with their preconceived opinions [19–22]. This behavioral propensity, inherently associating aligned advice with acceptance, may shapes advisors' expectation in the outcome of advice giving (i.e., whether their advice will be accepted or rejected), by adaptive reinforcement in everyday interactions [23,24]. Supporting this assumption, research has shown that advisors strategically gain acceptance from advisees by tailoring their confidence-expressing strategies to align with advisees' advice-taking habits [6,7], reflecting acquired knowledge of advisees' behavioral propensities. Recent research directly shows that individuals' willingness to provide advice reflects a complex interplay between informational and social factors [25]. People become more inclined to share their opinions as advice as their knowledge increases, and this tendency is further amplified when social rewards are introduced, namely, when they are informed that their advice and performance ratings will be visible to future advisees. This finding suggests that advice-giving, often seen as an information-driven behavior, can be shaped by anticipated feedback from advisees. This further proposes a striking yet plausible possibility that has been suggested by previous work [4,26]: advisors—typically seen as impartial and in control of influencing others—may also be swayed by social interests and, in turn, influenced by advisees, who are conventionally seen as the ones being influenced.

In summary, while evidence suggests the existence of alignment bias, direct investigations into this phenomenon and its emergence remain lacking. In this work, we uncovered the presence of alignment bias in social advice giving and systematically examined whether repeated feedback on advice shapes advice-giving tendencies, to provide an evolutionary perspective of the emergence of alignment bias. Across our studies, we simulated a real-time interaction context in which participants believed they were advising a human partner whose performance could be influenced by their advice and who would provide a subjective evaluation of their advice after interaction. In Study 1 ($n = 73$) and Study 2 ($n = 62$), participants were tasked with providing advice independently and after observing the advisee's opinions (either congruent or incongruent with participants' initial opinions) in two sequential sessions. In Study 1, we identified the existence of alignment bias and investigated whether the re-alignment towards advisees' incongruent opinions was contingent on their correctness. In Study 2, we investigated the generalizability of this bias under a performance-based incentive structure and examined whether advisors' re-alignment to advisees' opinions reflected a social basis (i.e., normative conformity), or merely reflected non-social processes and unconscious influences (e.g., opinion contagion [27] or anchoring effects [28]). In Study 3 ($n = 111$) and Study 4 ($n = 100$), immediate feedback on advice (acceptance or rejection) was presented to advisors in each trial, which was designed to reflect advisees' varying preferences for either aligned or misaligned advice. Combining behavioral measurements and computational models, we addressed whether and how advisors adapted their advice-giving tendencies to these varying preferences, reflecting their susceptibility to feedback on advice (Study 3). Moreover, we also examined whether these adaptations persisted when acceptance from advisees was scant (Study 4). Collectively, our research illuminated the intrinsic nature of alignment bias in social advice giving.

## 2. Study 1

In Study 1, participants were led to believe, through a deceptive instruction, that they would engage in real-time interaction with an advisee throughout both sessions of the task. Specifically, they were tasked to provide advice on investment decisions independently and after observing an advisee's opinions, in two separate sessions. We first validated the existence of alignment bias by investigating whether advice provided after observing advisees' opinions exhibited a greater degree of similarity with them compared to the baseline level (i.e., before observing advisees' opinions). Notably, we did not incorporate any feedback from advisees in order to capture alignment bias in its natural form that was not induced by additional information. Building on the existence of alignment bias, we continued to investigate the role of normative conformity in the emergence of alignment. To answer this question, we focused on whether advisors' re-alignment towards advisees' opinions was contingent on their accuracy [11].

### 2.1 Methods

**2.1.1 Ethics statements.** All experiments in this paper received approval from the Department of Psychology and Behavioral Sciences, Zhejiang University (No. 2024052), and were conducted in accordance with the Helsinki Declaration. All participants gave written informed consent before the study.

**2.1.2 Participants.** We selected an effect size of $d = 0.4$, following Cohen's (1988) convention for a medium effect [29], which we considered a reasonable benchmark given the limited prior evidence. To ensure adequate sensitivity while keeping the sample size feasible, we set the power level at 85% (above the conventional 80% threshold). We determined the sample size through a power analysis conducted using G*Power 3.1, which indicated that a sample of at least 59 participants was required. To enhance statistical power, we opted to oversample to ensure a minimum of 70 participants in each experimental group. To achieve this, we recruited 80 participants to ensure adequate sample size after potential participant exclusions. After excluding participants ($n = 7$) not passing attention checks, 73 participants (mean age: 22.23 years, range: 18–29 years; 42 females) were included in the final analyses.

**2.1.3 Experimental task.** The advice-giving task was adapted from a risk investment game [19], where players make judgements on the prices of estates properties, and place wager on judgements (players win their wagers on correct judgements as incentive). Before the experiment, two participants entered the lab at the same time and received the study instructions together. In this verbal instruction, they were informed that each participant would be assigned either the role of advisor or advisee, to convince them that they were indeed playing a real-time interaction game. The advisee's bonus depends directly on the number of wagers won through their decisions, with the advisor's role being to assist the advisee in maximizing winnings by providing advice. Both players completed two sessions involving identical stimuli, with the later session serving a second chance to revise decisions or advice (this was designed to conceal the true purpose of the manipulation). During the game, neither players would receive feedback on their accuracy or performance. After this verbal introduction, participants entered separate booths and received role-specific instructions. In fact, all participants were assigned to the advisor role and read the cover story as below.

'In both sessions, the advisee will first provide his/her initial opinion in each trial and then receive your advice immediately. They can decide whether to accept it as their decision in that trial (or reject it but maintaining the initial opinion). In Session 1, you are required to provide your advice to the advisee right after the display of an estate photo and a price. In Session 2, you will be additionally presented with advisee's opinion recorded in Session 1 in each trial. After that, you provide your advice. Before being informed of final performance in the task, the advisee would provide a subjective evaluation of your advice on a scale from 1 to 10, which was directly converted into your payment bonus (1 to 10 CNY).'

In fact, the subjective evaluations from advisees were randomly generated. Notably, we explicitly informed participants that neither the advisor nor the advisee would receive performance-related feedback. This simulated real-world advice-giving contexts in which the accuracy of advice and the consequence of its adoption are not immediately verifiable. It also mitigated the possibility that alignment merely reflected an attempt to avoid evaluative penalty for misaligned advice

that proved to be incorrect. Therefore, the perception of real-time interaction with a human partner, combined with role framing that introduced reward interdependence between advisor and advisee, helped construct a situated social connection in our experiments. These manipulations ensured that the advice-giving behavior observed in our studies reflected a social process, rather than a purely economic decision as in non-social situations.

In summary, participants were tasked with providing advice in both *non-social* and *social* scenarios sequentially (Fig 1a), each consisting of 66 trials using identical stimuli. In the non-social scenario (Session 1), participants gave advice independently to the interacting advisee right after each stimulus (an estate and its possible price) was displayed (i.e., advising without observing the opinion from the advisee). In the social scenario (Session 2), participants observed all the stimuli again and provided advice after displayed with the interacting advisee's initial opinions (both judgements and investments) on the stimuli, except for 20% of trials (14 trials) where advisees' opinions were masked (identical to the *non-social* scenario). The inclusion of masked trials within the social advice-giving scenario was designed to investigate how advisees' behavioral preferences (specifically, risk preference) shaped advice giving while isolating the trial-by-trial influence of opinions from advisees (see **2.1.6** for details). Note that, we fixed the order of Session 1 and 2 (instead of counterbalancing) for two reasons: (1) participant data from Session 1 were required to generate the alleged advisees' opinions in Session 2; (2) having seen advisees' opinions in Session 2 could potentially jeopardize the independence of non-social advice given in Session 1. Both would hamper the experimental design and clear interpretation of the results.

Unbeknownst to participants, advisees they interacted with were alleged. In fact, advisees' opinions were preprogrammed (therefore deceptive approaches had been used in the experiments). Advisees' judgements were generated according to participant's non-social judgements in Session 1 (hence, the non-social session had to be completed before the social session), ensuring that in half of the trials wherein advisees' opinions were presented, advisees held

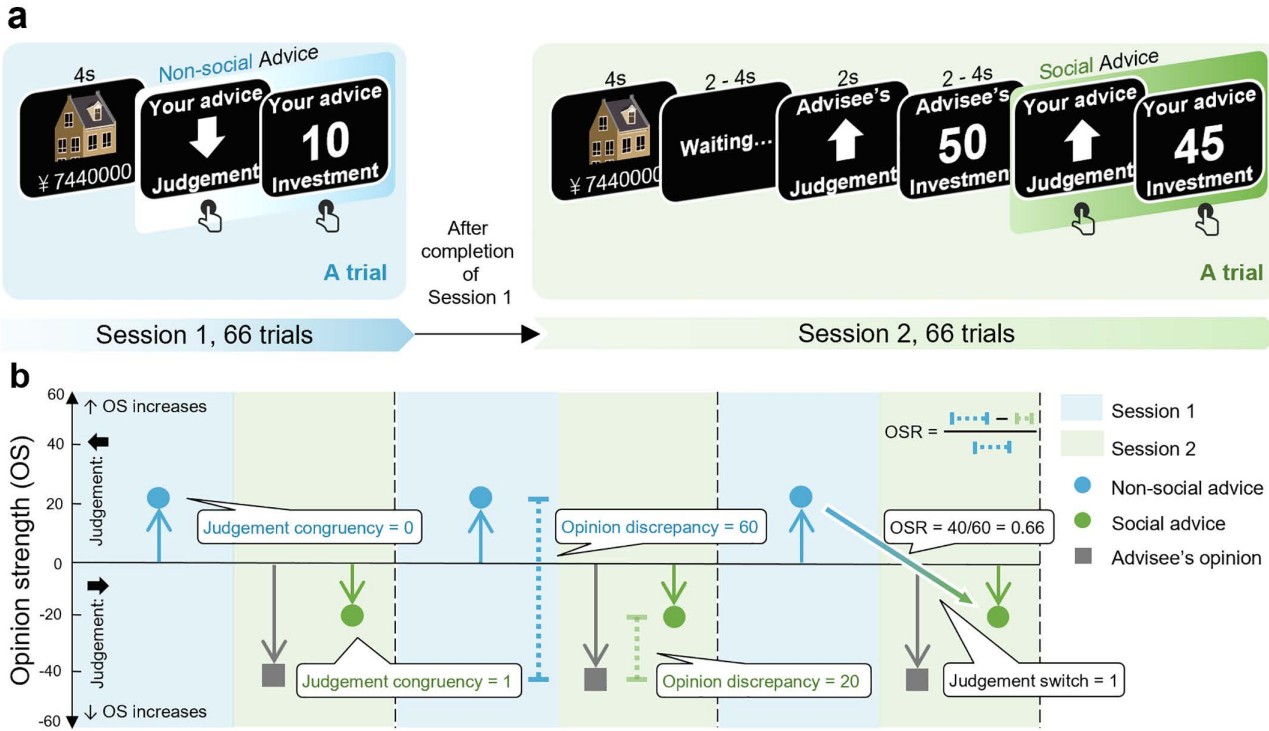

**Fig 1. Experimental task in Study 1 and conceptual schematics of key behavioral measurements. (a)** Task design. **(b)** Conceptual schematics of the key behavioral measurements on the alignment bias.

incongruent judgements with the advice given by participants in Session 1 (the other half of the trials were hence congruent). Advisees' investments were all artificially generated from normal distributions, *Normal*(10,10) or *Normal*(50,10), distinguish their risk-avoiding or risk-seeking risk preferences as a between-subject variable. These distributions were designed to contrast with participants' typical investments (around 30) in the non-social session, thereby allowing for adaptations in the social session. In other words, this setup aimed to investigate whether advisors gradually aligned their advised investments to align with advisees' risk preferences on the basis of observations of advisees' decision.

**2.1.4 Stimuli.** To create the stimuli, 66 real estate properties were selected from a domestic estate trading website. The stimulus price was either 20% higher or lower than the real price of a property, and all values were rounded to the nearest whole number to ensure consistency with common pricing conventions. These stimuli were randomly divided into 3 subsets and assigned to different conditions (advisee's judgement was congruent, incongruent with participants' non-social judgement, or was not presented) with counterbalance.

A pretest was conducted on the stimuli subsets (see S1 Text for details). No significant differences were detected between these stimuli subsets in both judgement confidence ($F(2,63) = 0.05$, $p = 0.95$) and familiarity ($F(2,63) = 0.71$, $p = 0.49$).

**2.1.5 Procedures.** After consent to participation, participants were required to read the task instructions and underwent a brief quiz to ensure they comprehensively understood the task requirements. Following this, participants engaged in a two-session advice-giving task. Upon completion of the task, participants were debriefed about the deceptive manipulations and received monetary compensation consisting of a base payment (30 CNY) and a bonus (Study 1: 5 CNY; Studies 2–4: 1–10 CNY, based on performance).

**2.1.6 Design, measurements and analytical plans.** We adopted a within-subject design in this study to investigate the alignment bias, by assessing inter-session differences between advice given without (Session 1, non-social scenario) and after observing advisees' opinions (Session 2, social scenario). Before conducting data analysis, we excluded trials where the value of the examined variable was missing, or where the value exceeded the constraint scales (e.g., investments magnitude outside the range of 1–60). Under the consideration of the fractional property, we incorporated additional exclusion criteria for OSR (see details in the next paragraph). After data exclusions, we constructed linear mixed-effect models to conduct statistical analysis using the lme4 package [30] in R (Version 4.3.1).

Generalized linear mixed-effect models were employed when the dependent variable was binary (e.g., judgement congruency or judgement re-alignment) using the *glmer*() function, assuming a binomial distribution for the response and reporting fixed-effect statistics as *z*-values. Linear mixed-effect models were used when the dependent variable was continuous (e.g., opinion congruency or OSR) using the *lmer*() function, with fixed-effect statistics reported as *t*-values along with the corresponding degrees of freedom. The lmerTest package was used to perform significance tests on the parameters yielded by the model [31]. Post-hoc comparisons were performed using the emmeans package [32]. Participants and stimuli (i.e., estate photos) were modeled as random intercepts across all analyses. All variables were mean-centered. We used the default 'nloptwrap' parameter optimization method to fit the models. The specifications of all mixed effect models used in the present study are provided in S2 Text. For statistical analysis involving only one observation per participant (e.g., individual-level computational parameters), we employed conventional statistical approaches using Jamovi computer software [33].

***Does alignment bias exist in social advice giving?*** To depict the alignment bias, two measurements—judgement congruency and opinion discrepancy—were computed and compared between Session 1 and Session 2 (**Fig 1b**). These measurements quantified the similarity between advice and the opinions of advisees. Changes in these measurements from Session 1 to Session 2 reflected how participants adjusted their advice to align with advisees' opinions, facilitating the emergence of alignment bias. Judgement congruency was coded as 0 or 1, indicating whether the participant's advised judgement was incongruent (0) or congruent (1) with the advisee's judgement regarding the same stimulus (**Fig 1b**). Opinion discrepancy captured the fine-grained difference between advice and advisees' opinions (**Fig 1b**), calculated

as the distance between them on the Opinion Strength (OS) scale (**Fig 1b**). On this scale, the sign of the value represents the judgement type ('+' for 'higher' and '−' for 'lower'), while the magnitude reflects the level of confidence in the judgement (measured by investment magnitude). Linear mixed-effect models were conducted to examine the inter-session differences of these measurements.

In addition to investigating the presence of alignment bias towards advisees' opinions, we further explored if advisors implicitly internalized and aligned towards the behavioral preferences of their advisees, revealing a subtle aspect of alignment bias. Specifically, we compared advisors' investment magnitude between Session 2 and Session 1 in trials where advisees' opinions were masked (to isolate the influence of trial-by-trial opinions from the advisee) and examined whether the direction of this effect varied depending on the conditions of advisees' risk preferences (risk-avoiding vs. risk-seeking).

***Does the alignment bias emerge from advisors' conformity to advisees' opinions?*** The alignment bias arises from advisors' inclination to re-align with advisees' opinions, which could be manifested as a greater likelihood of changing their mind (measured by judgement switch, **Fig 1b**) and larger opinion shifts (measured by OSR, **Fig 1b**) towards advisees' opinions that were incongruent (compared to congruent) with their initial opinions (i.e., non-social advice). Specifically, the judgement switch was coded as 1 or 0, depending on whether advisors switched their initial judgement (from Session 1) in Session 2. Opinion shift rate (OSR) quantified the degree of adjustment towards advisees' opinions. Following established measures of social influence (e.g., others' advice [20] or opinions [34]) on individuals' decision-making, OSR in each trial was computed as the intensity of adjustment (i.e., the distance between the opinion strength in Session 1 and the opinion strength in Session 2) divided by the baseline opinion discrepancy (i.e., the opinion discrepancy in Session 1). Specifically, an OSR larger than 0 indicates advice adjusted towards the advisee's opinion, while an OSR smaller than 0 indicates advice deviated from the advisee's opinion. For analyses involving Opinion Shift Rate (OSR) as a variable, we excluded trials involving inexplicable OSR situations. The first type of inexplicable situations occurred when the advisor's opinion strength ($OS_t^{ns}$, indicating the advisors' opinion strength in the non-social scenario; $OS_t^s$, indicating the advisors' opinion strength in the social scenario) shifted towards the advisee's opinion strength ($OS_t^{as}$) (i.e., OSR > 0), however, the judgement ended up incongruent with the advisee's judgement (e.g., $OS_t^{ns}$ = -40, $OS_t^{as}$ = -5, and $OS_t^s$ = 10). This situation posed a challenge in defining whether it constituted a 'shift towards' or a 'deviate from' the advisee's opinion. The second type of inexplicable situations occurred when $OS_t^{ns}$ = $OS_t^{as}$, rendering the $OSR_t$ value meaningless. In addition, we excluded extreme values of OSR (beyond $M \pm 3SD$ of each participant) from the analyses involving OSR as a variable.

A key of assessing the conformity-driven nature of alignment bias was whether advisors' tendency to re-align with advisees' opinions was contingent on the accuracy of those opinions. To examine whether the likelihood of re-alignment was influenced by the accuracy of advisees' opinions, we conducted two analyses. First, we included all trials to assess its overall effect. Second, we focused specifically on the subset of trials where participants had indicated higher confidence in their initial judgements than their advisees did, to test whether the effect persisted. Accuracy was coded according to the ground truth—specifically, whether the stimulus price was higher or lower than the actual price. Participants were expected to exhibit a tendency of judgement re-alignment—increased judgement switch in the incongruent (compared to the congruent) condition—in both conditions of advisees' judgement accuracy (correct/ incorrect), i.e., an unselective re-alignment pattern.

However, given participants' limited knowledge of property evaluation, the pattern of unselective re-alignment might merely reflect their over-reliance on an irrelevant cue—specifically, advisees' confidence in their judgements (indicated by investment magnitude, which was artificially generated and thus not indicative of the actual accuracy)—to calibrate their advice [35,36]. By this assumption, judgement re-alignment was expected to be more strongly predicted by advisees' confidence in their judgements, compared to their accuracy. Therefore, we constructed linear mixed-effect models to investigate the influence of advisees' judgement accuracy and their confidence on judgement re-alignment, controlling for advisors' confidence in their initial judgements (provided in Session 1).

Finally, we explored if the re-alignment pattern could be extended to a fine-grained scale. We constructed a similar linear mixed-effect model on OSR to examine the influences of judgement congruency and advisees' judgement accuracy. The larger OSR for incongruent opinions could be driven by the broader range of potential variation. For example, when judgements were initially congruent between advice and advisees' opinions, advice could at most shift over half the Opinion Strength scale. By contrast, when judgements were incongruent, advice could potentially shift over the entire scale. This introduced a potential confounding influence of baseline opinion discrepancy. To address this conjecture, we conducted an additional analysis on OSR, controlling for the influence of the baseline opinion discrepancy.

## 2.2 Results

**2.2.1 Advisors aligned their advice towards advisees' opinions.** Participants demonstrated an alignment bias towards advisees' opinions (**Fig 2a**), as evidenced by their increased judgement congruency with their advisees' opinions in the social compared to non-social scenario ($\beta = 0.64$, $z = 13.43$, 95% CI = [0.55, 0.74], $p < 0.001$; **Fig 2b**). The alignment bias was also evident on the reduced opinion discrepancy with their advisees in the social compared to non-social scenario ($\beta = -12.62$, $t(7708.1) = -22.15$, 95% CI = [-13.73, -11.50], $p < 0.001$; **Fig 2c**). The analysis of OSR revealed that participants shifted towards (OSR > 0) advisees' opinions in most of the trials (**Fig 2d**). Notably, participants were observed to frequently over-weigh advisees' opinions than their own (OSR > 0.5) and even excessively agreed with advisees' opinions (OSR > 1).

Advisors not only aligned with the opinions of their advisees but also exhibited a subtle tendency to gravitate toward advisees' risk preferences during the interaction. Compared to providing advice independently, participants systematically adjusted their overall investment magnitude in response to advisees' risk preferences—decreasing when interacting with risk-averse advisees ($\beta = -10.27$, $t(1657) = -11.64$, 95% CI = [-12.00, -8.54], $p < 0.001$) and increasing when interacting with risk-seeking advisees ($\beta = 3.38$, $t(1657) = 3.78$, 95% CI = [1.62, 5.14], $p < 0.001$).

**2.2.2 Advisors exhibited an unselective inclination to re-align with advisees' opinions.** Next, we investigated whether the tendency of advisors to re-align with advisees' judgements depended on their actual accuracy. (**Fig 2e**). The results showed that, advisors were more likely to switch their judgements when their initial judgements (i.e., non-social advice) were incongruent with advisees' judgements, as opposed to congruent ($\beta = 1.81$, $z = 19.48$, 95% CI = [1.62, 1.99], $p < 0.001$). The interaction effect of judgement congruency × advisees' judgement accuracy ($\beta = 0.91$, $z = 4.99$, 95% CI = [0.55, 1.27], $p < 0.001$) revealed that participants were more likely to re-align their judgements with correct judgements from advisees ($\beta = 2.26$, $z = 17.01$, 95% CI = [2.00, 2.52], $p < 0.001$), while still showing a notable inclination to re-align with incorrect judgements ($\beta = 1.35$, $z = 10.61$, 95% CI = [1.10, 1.60], $p < 0.001$). Moreover, in trials where participants expressed higher confidence in their initial judgements than their advisees did, the unselective tendency to re-align persisted (incorrect: $\beta = 1.19$, $z = 2.57$, 95% CI = [0.28, 2.10], $p = 0.01$; correct: $\beta = 3.39$, $z = 6.24$, 95% CI = [2.32, 4.45], $p < 0.001$).

However, one might be concerned that the observed unselective re-alignment could simply result from participants' over-reliance on advisees' confidence (which was not actually indicative of their accuracy). This could also lead to an unselective re-alignment pattern that lacked discrimination of advisees' accuracy. Contrary to this conjecture, judgement re-alignment was more strongly influenced by advisees' actual judgement accuracy ($\beta = 0.40$, $z = 3.54$, 95% CI = [0.18, 0.62], $p < 0.001$) compared to their expressed confidence ($\beta = 0.008$, $z = 1.89$, 95% CI = [-0.0003, 0.02], $p = 0.06$). Additionally, advisors' confidence in their non-social judgement also negatively predicted judgement re-alignment ($\beta = -0.04$, $z = -10.92$, 95% CI = [-0.05, -0.04], $p < 0.001$), providing further evidence against the possibility of advisors' over-reliance on advisees' confidence leading to the observation of unselective re-alignment.

Echoing with the findings on judgement switch, advisors exhibited a larger OSR towards advisees' incongruent (than congruent) opinions ($\beta = 0.13$, $t(3593) = 3.98$, 95% CI = [0.07, 0.19], $p < 0.001$) and their correct (than incorrect) opinions ($\beta = 0.09$, $t(3618) = 2.72$, 95% CI = [0.03, 0.15], $p = 0.007$) (**Fig 2f**). The effect of interaction was non-significant ($\beta = 0.04$,

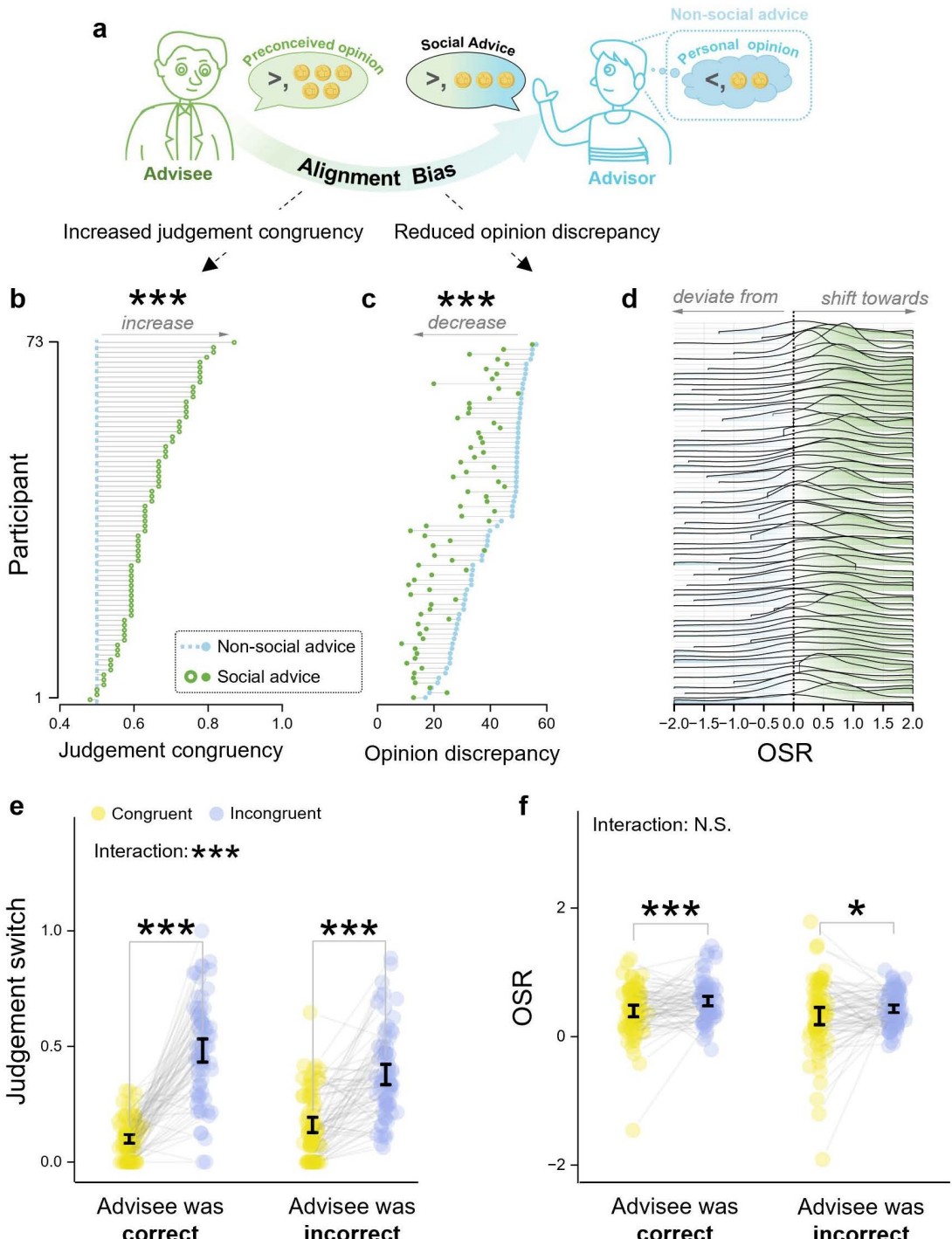

**Fig 2. Advisors tended to align their advice with advisee's opinions. (a)** The schematic of alignment bias in social advice giving. **(b)** Judgement congruency and, **(c)** opinion discrepancy with advisee's opinion in non-social vs. social advice-giving scenario of each participant. **(d)** The frequency of opinion shift rate (OSR) among all trials of each participant. **(e)** Advisors exhibited a larger inclination to switch their judgements when encountering advisees' incongruent (than congruent) opinions. **(f)** Advisors exhibited larger OSRs towards advisees' incongruent (than congruent) opinions. Dots represent individual participants' data; error bars represent the 95% confidence intervals (CIs).

$t(3644) = 0.67$, 95% CI = [-0.09, 0.17], $p = 0.50$). These results remained consistent when controlling for the influence of baseline opinion discrepancy (judgement congruency: $\beta = 0.13$, $t(3592) = 3.99$, 95% CI = [0.07, 0.19], $p < 0.001$; advisees' judgement accuracy: $\beta = 0.09$, $t(3617) = 2.73$, 95% CI = [0.03, 0.15], $p = 0.01$; judgement congruency × advisees' judgement accuracy: $\beta = 0.05$, $t(3643) = 0.76$, 95% CI = [-0.08, 0.18], $p = 0.45$). The influence of the baseline opinion discrepancy on OSR was non-significant ($\beta = 0.001$, $t(2988) = 0.98$, 95% CI = [-0.001, 0.03], $p = 0.33$).

## 2.3 Discussion

Study 1 uncovered the presence of alignment bias in social advice-giving. Specifically, we observed that advisors tended to adjust their advice towards advisees' opinions. Additionally, advisors even exhibited a tendency of gravitating their advised investments to the risk preferences of advisees, in the circumstances where advisees' opinions were not presented. Further analyses indicated that advisors displayed an unselective tendency to re-align with advisees' opinions, shifting initially incongruent advice to align with those opinions, even when they were incorrect, demonstrating the conformity-driven nature of the alignment bias. Importantly, our data also showed that the unselective re-alignment to advisees' opinions could not be simply attributed to advisors' over-reliance on irrelevant information which was not indicative of the accuracy of advisees' opinions.

## 3. Study 2

Evidence from Study 1 provided pioneering insights into the conformity-driven nature of the alignment bias, demonstrated by an unselective inclination to re-align with advisees' incongruent opinions. However, it remained unclear whether this conformity reflected a social basis or instead arose from non-social processes (e.g., information seeking, copying strategies) and unconscious influences (e.g., opinion contagion [27] or anchoring effects [28]). Additionally, one might be concerned that the unselective re-alignment with advisees' judgements, regardless of their accuracy, could rather simply reflect the absence of direct motivation for participants to discriminate incorrect opinions from advisees and calibrate their advice.

To disentangle alignment bias from non-social processes and unconscious influences, we introduced an additional session (Session 3) in Study 2, where participants gave advice independently again, while being informed that their advice would be provided either to the *same* advisee from Session 2 or a *new* advisee. Participants advising a new advisee were expected to exhibit a greater decline in re-alignment with the advisee's opinions from Session 2 compared to the participants in the same-advisee condition. Conversely, if alignment bias was solely due to non-social processes or unconscious influences, no difference in the decline of re-alignment between these conditions would be anticipated.

Additionally, to address the possibility that re-aligning with advisees' opinions unselectively served merely as a strategy to conserve effort on advice calibration, as well as to further examine the generalizability of alignment bias and its intrinsic motivational basis, we replaced the evaluation-based incentive structure used in Study 1 with a performance-based structure in Study 2. In this design, participants' bonuses were directly linked to the quality of their advice, thereby eliminating any economic motivations to pursue unselective re-alignment.

### 3.1 Methods

**3.1.1 Participants.** Given the pronounced effects observed for alignment bias in Study 1, we relaxed the criterion of a strict minimum sample size of 70 but ensured a sample size no smaller than 58, as determined by the power analysis ($d = 0.4$, with a Type I error of .05 and a power level of .85). After excluding participants ($n = 2$) not passing attention checks, 62 participants (mean age: 23.03 years, range: 18–32 years; 39 females) were included in the final analysis. No participants expressed suspicion regarding the authenticity of the advisee in the post-task survey, and thus no exclusion was made on this basis.

**3.1.2 Experimental task.** Different from Study 1, four participants entered the lab at the same time, to make them believe that there were two advisor-advisee pairs (hence two potential advisees). This was designed to enable the manipulation regarding a "new" advisee in Session 3. Participants were jointly introduced to the study by the experimenter and read the cover story as we used in Study 1, including the information that they would receive the advisee's subjective evaluation at the end of the task. However, they were informed that their bonus would be calculated based on the trial-by-trial performance of advice (i.e., accuracy × wager, with accuracy coded as –1 for incorrect and 1 for correct). The overall performance score will be linearly rescaled to a 0–10 range and converted to payment bonus (1–10 CNY), which was added to the base payment.

Additionally, in Study 2, the task comprised 3 sessions. Session 1 and 2 were consistent with Study 1. In Session 3, which included the same number of trials and identical stimuli as Sessions 1 and 2, participants again provided advice independently as in Session 1. However, before entering this session, participants were informed that their advice would be provided either to a new advisee or to the same advisee with whom they had previously interacted in Session 2. Notably, participants were not given any instructions regarding Session 3, nor were they informed about the possibility of encountering the same advisees from Session 2 again during the earlier stages of the experiment.

**3.1.3 Post-task survey.** To strengthen the manipulation check regarding participants' perception of the interactive context (that is, a belief that they were interacting with a human advisee), a post-task survey was introduced.

Participants were first asked to report on their belief of playing the role of advisors by rating on their level of agreement with the statements below on a scale of 1 (= "strongly disagree") to 9 (= "strongly agree"):

1. 'I am aware that I played the role of an advisor.'

2. 'My advice was provided independently.'

3. 'My advice benefited my advisee.'

4. 'I made my advice after careful consideration.'

5. 'I mostly relied on my advisee's opinions to provide advice.'

Participants who rated lower than 5 on the first four items or higher than 5 on the fifth item were excluded from the final analysis, as this indicated miscomprehension of the role of advisors.

Second, upon completion of the experiments, participants were asked to share their subjective experiences during the experiments, in response to experimenter's open-end asking, 'Did you have any thoughts or impressions about the advisee player you were interacting with during the task?'. Notably, Participants were not informed in advance that they would be asked this question, and we made no indication that their responses would have any influence, in order to minimize response bias and encourage honest answers. Those who indicated that they perceived the advisees as fictitious would also be excluded; however, no participants met this criterion, and therefore none were excluded on this basis.

**3.1.4 Design, measurements, and analytical plans.** We employed the same design and analytical plans as in Study 1 to replicate the findings based on data from Sessions 1 and 2.

Data from Session 3 were analyzed to investigate whether the conformity to advisees' opinions involved social motivations (i.e., normative conformity) or instead reflected non-social processes and unconscious influence. Specifically, we employed a linear mixed-effects model to test whether the decrease in judgement congruency with advisees' Session 2 opinions, from Session 2 to Session 3, differed between participants advising the same advisee versus a new advisee. To further assess whether this difference reflected a decreased tendency to maintain re-alignment, we conducted a follow-up analysis focusing on trials where participants had previously re-aligned with the advisees' judgements in Session 2, comparing the advisee conditions.

Additionally, the observed difference in re-alignment maintenance might alternatively reflect a greater motivation to maintain behavioral consistency when interacting with the same advisee. To examine this possibility, we further constructed a linear mixed-effects model comparing the likelihood of repeating the same judgement from Session 2 across advisee conditions.

### 3.2 Results

**3.2.1 Replications on the alignment bias and the unselective re-alignment pattern.** Participants exhibited alignment bias in social advice-giving scenario, as evidenced by increased judgement congruency ($\beta = 0.50$, $z = 9.61$, 95% CI = [0.40, 0.60], $p < 0.001$) and decreased opinion discrepancy ($\beta = -8.13$, $t(6507.2) = -13.60$, 95% CI = [-9.31, -6.96], $p < 0.001$). Additionally, advisors' investment magnitude was also found to gravitate towards their advisees' risk preferences, though we only observed a trend within interactions with risk-seeking advisees (risk-avoiding advisee: $\beta = -5.51$, $t(1392) = -6.04$, 95% CI = [-7.30, -3.72], $p < 0.001$; risk-seeking advisee: $\beta = 1.44$, $t(1392) = 1.48$, 95% CI = [-3.36, 0.48], $p = 0.14$). The generalizability of alignment bias was further demonstrated by advisors shifting towards (OSR > 0) advisees' opinions in most trials (S1 Fig).

Consistent with Study 1, participants exhibited an unselective inclination to re-align with advisees' judgements. The results showed that advisors were more likely to switch their judgements when their non-social advice was incongruent with advisees' judgements ($\beta = 1.34$, $z = 14.49$, 95% CI = [1.16, 1.52], $p < 0.001$). A larger interaction effect of judgement congruency × advisees' judgement accuracy was found compared to Study 1 ($\beta = 1.03$, $z = 5.54$, 95% CI = [0.67, 1.39], $p < 0.001$), suggesting participants' enhanced motivation of advice calibration when accuracy was directly incentivized. However, the inclination of re-alignment persisted both when advisee's judgement was correct ($\beta = 1.86$, $z = 13.79$, 95% CI = [1.59, 2.12], $p < 0.001$) and when it was incorrect ($\beta = 0.83$, $z = 6.49$, 95% CI = [0.58, 1.08], $p < 0.001$). Moreover, in trials where participants expressed higher confidence in their initial judgements than their advisee did, the unselective re-alignment remained evident (incorrect: $\beta = 0.71$, $z = 3.76$, 95% CI = [0.34, 1.08], $p < 0.001$; correct: $\beta = 1.69$, $z = 8.59$, 95% CI = [1.30, 2.08], $p < 0.001$).

Furthermore, the unselective re-alignment pattern was also manifested in opinion shift rate (OSR). Advisors exhibited a larger OSR towards advisees' incongruent opinions compared to congruent opinions ($\beta = 0.18$, $t(1477.2) = 4.31$, 95% CI = [0.10, 0.26], $p < 0.001$) and towards correct opinions compared to incorrect ones ($\beta = 0.16$, $t(2782.5) = 3.81$, 95% CI = [0.03, 0.92], $p < 0.001$). The effect of interaction was non-significant ($\beta = -0.02$, $t(815.8) = -0.25$, 95% CI = [-0.19, 0.14], $p = 0.81$). These results remained consistent when we controlled for baseline opinion discrepancy (judgement congruency: $\beta = 0.18$, $t(1407) = 4.32$, 95% CI = [0.10, 0.26], $p < 0.001$; advisees' judgement accuracy: $\beta = 0.16$, $t(2753) = 3.81$, 95% CI = [0.08, 0.24], $p < 0.001$; judgement congruency × advisees' judgement accuracy: $\beta = -0.02$, $t(795.9) = -0.26$, 95% CI = [-0.19, 0.14], $p = 0.79$). The influence of the baseline opinion discrepancy was also non-significant ($\beta = -0.001$, $t(1069) = -0.94$, 95% CI = [-0.004, 0.001], $p = 0.35$).

**3.2.2 The conformity to advisees' opinions reflects a social basis.** Next, we examined the normative features of the conformity to advisees' opinions by analyzing advice-giving patterns in Session 3. Consistent with our hypothesis, from Session 2 to Session 3, participants in the new-advisee condition exhibited a greater decrease in judgement congruency compared to those in the same-advisee condition ($\beta = 0.26$, $z = 2.53$, 95% CI = [0.06, 0.46], $p = 0.01$). Moreover, the re-align judgements observed in Session 2 were less likely to be maintained by participants in Session 3, in the new-advisee condition than by those in the same-advisee condition ($\beta = -0.49$, $z = -2.48$, 95% CI = [-0.87, -0.10], $p = 0.01$). Furthermore, this between-condition difference could not be attributed to a motivation to preserve a self-consistent image by maintaining previous judgements, as participants in the same-advisee condition did not show greater cross-session consistency in their advised judgements compared to those in the new-advisee condition ($\beta = 0.20$, $z = 1.40$, 95% CI = [-0.08, 0.48], $p = 0.16$).

### 3.3 Discussion

The replications in Study 2 further reinforced the findings from Study 1 regarding the existence of alignment bias and additionally ruled out the possibility that advisors' conformity to advisees' opinions, particularly incorrect ones, stemmed solely

from a lack of motivation to calibrate their advice. Importantly, the maintenance of re-alignment with advisees' opinions diminished when participants switched to provide advice to a new advisee, compared to the advisee they had previously interacted with. This finding ruled out the possibility that advisors' re-alignment to advisees' opinions merely resulted from information-seeking behaviors or having influenced by advisees' opinions unconsciously, instead emphasizing its normative nature.

## 4. Study 3

The combined results from Study 1 and Study 2 provided substantial evidence that normative conformity underpinned the emergence of alignment bias. To examine the potential value-based learning account of this conformity, we incorporated immediate feedback on advice (acceptance/rejection from advisees) into the social advice-giving scenario (Session 2) and arranged it into a probabilistic reversal design [37], manifested as advisees' volatile preferences for taking either aligned or misaligned advice. In other words, the probability of receiving acceptance feedback was influenced by whether the advice matched advisee's periodical preference. With this setup, advisors were expected to shift from invariably aligning with advisees' opinions (as in Study 1 and 2) to adapting their advice-giving tendencies with advisees' preferences based on the feedback information.

### 4.1 Methods

**4.1.1 Participants.** To detect a medium effect ($f = 0.2$) [38] in the ANOVA analysis examining the inter-phase adaptations of advice-giving tendencies across two presenting orders, with a Type I error rate of .05 and a power level of .85, at least 78 participants were required. To ensure robust computational model estimations, 120 participants were recruited in total. After excluding participants who failed attention checks ($n = 5$) and those who demonstrated misunderstandings of the task in the post-task questionnaire regarding their belief of playing the role of advisors ($n = 4$), the final sample consisted of 111 participants (mean age: 21.90 years, range: 18–30 years; 69 females). No participants expressed suspicion regarding the authenticity of the advisee in the post-task survey, and thus no exclusion was made on this basis.

**4.1.2 Experimental task.** The experiment settings were mainly consistent with Study 2, except that feedback on advice (acceptance/rejection) was disclosed in each trial after advisors submitted their advice in Session 2. According to the cover story, the advisee could either accept the advice, making it their final decision, or reject it and retain their initial opinion. Participants were explicitly informed that neither they nor the advisees had access to any information about judgement accuracy, and that their payment depended solely on their own task performance, not on the advisee's evaluations. As a result, participants understood that the advisee's feedback functioned purely as a social signal, rather than as an indicator of performance or monetary reward. Feedback was administered into a probabilistic reversal design [37], reflecting advisees' preferences for either aligned or misaligned advice (**Fig 3a**). Session 2 was truncated into three phases, each consisting of 22 trials, and advisees' preferences fluctuated between phases (**Fig 3a**). In each phase, advisees' preferences for a specific type of advice (we referred this to 'phased preference') was presented by a higher probability of acceptance, $P$(accept | advice type), compared to the other advice type. For example, as advisees exhibited a phased preference for aligned advice, aligned advice would be accepted at a higher rate (70%) than misaligned advice (30%), and vice versa. We presented feedback without displaying the advice type in that trial, to capture value-based learning in its natural form, allowing advisors to spontaneously notice and adapt to the contingencies between advice type and feedback.

To emulate typical everyday advice-interaction, mild advisees were simulated, who moderately accepted advice throughout the task, with an overall acceptance rate of 50%. Therefore, the acceptance rate for each type of advice was distributed across three levels: 30%, 50%, and 70%. In the initial phase, both aligned and misaligned advice had the same acceptance rate at 50%. This phase was designed to serve as a control condition for measuring advisors'

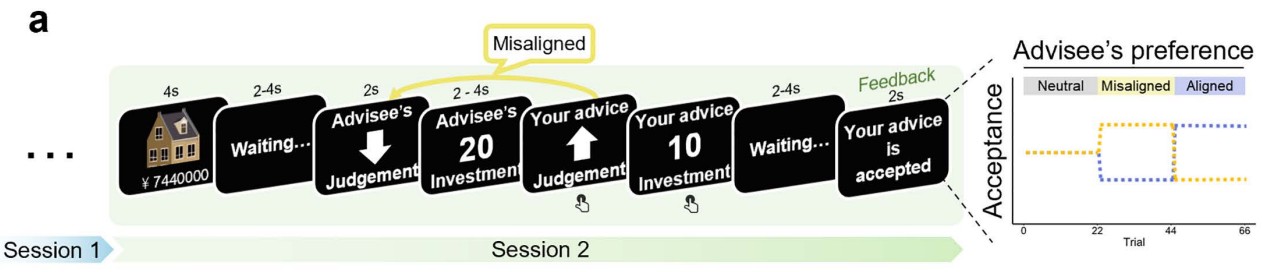

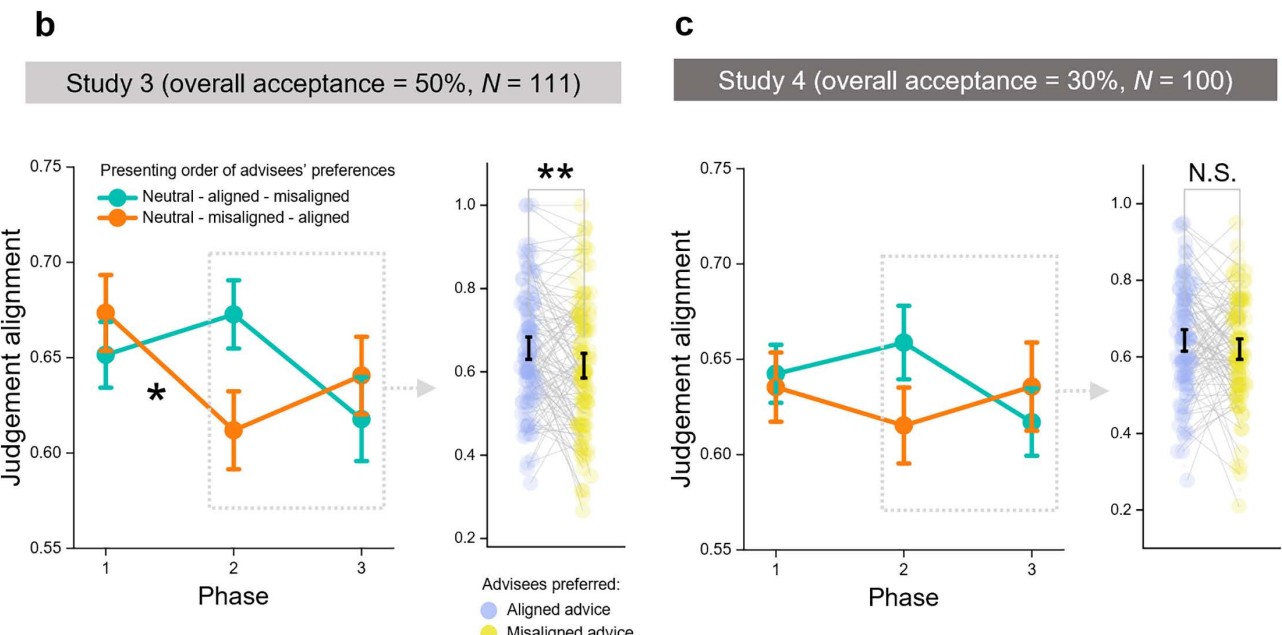

**Fig 3. Experimental tasks in Study 3 and Study 4 and advice-giving tendencies adapted with advisees' varying preferences. (a)** Experimental designs in Study 3 ($n$ = 111) and Study 4 ($n$ = 100). **(b)-(c)** Judgement alignment with advisee's opinions in Study 3 and Study 4, presented in a phase-by-phase format (left panel) and as a comparison between advisees' preference for aligned vs. misaligned advice, collapsing ordering conditions (right panel). Dots represent means; error bars represent the SEs in the left panel. Scatter represents individual data in the right panel.

adaptation to advisees' varying preferences across phases. In the subsequent two phases, the order in which 'preference for aligned advice' and 'preference for misaligned advice' were presented was counterbalanced across participants, enabling a comparison of advisors' advice-giving tendencies while pooling the ordering condition to maximize statistical power.

**4.1.3 Model-free analyses.** To depict the advice-giving tendencies that adapting with advisees' phased preferences, we first constructed a linear mixed-effect model to examine whether the extent to which participants provided aligned advice (measured by judgement alignment) varied between phases in both presenting orders of advisees' preferences ('neutral-aligned-misaligned' vs. 'neutral-misaligned-aligned'). To maximize statistical power in detecting potential effects, we further constructed a linear mixed-effect model on judgement alignment, focusing on the phases where advisees preferred aligned vs. misaligned advice, while pooling the ordering conditions (the 'neutral' phase was excluded as its presenting order was not counterbalanced).

Moreover, we constructed a linear mixed-effect model to examine the relationship between judgement alignment (aligned advice = 1, misaligned advice = 0) and task process (i.e., trial), in order to investigate whether the alignment bias overall diminished with feedback acquisition.

**4.1.4 Computational modeling.** To capture advisors' subtle adaptations based on feedback from their advisees, we employed a computational modeling approach to parse the trial-by-trial dynamics of the advice-giving behaviors. Three categories of computational models were constructed (**Table 1**). These models separately assumed whether and how advisors adjusted their advice in response to advisees' feedback (see **4.2.1** for details).

The 'baseline' category (M1) posited that, advisors solely depended on their own opinions to provide advice without considering any social information. This model served as the baseline for model comparison. The 'social bias' category (M2-M4) proposed that, alignment bias reflects a tendency to maintain interpersonal coherence or to avoid taking responsibility for misadvising. Thus, advisors maintained a tendency to align towards advisees' opinions, which were not shaped by advisees' preferences. This category also accommodated the possibility that alignment bias simply reflected anchoring effects or opinion contagion—that is, advisors being permeated by advisees' opinions persistently. The 'social learning' category (M5-M7) posited that alignment bias can be a behavioral phenomenon emerge from a general value-based learning mechanism. Hence, advisors' pre-existing tendency to align with advisees' opinions can be shaped by feedback through a reinforcement learning process, resulting in behavioral adaptations that manifest as greater accordance between advice-giving tendencies and advisees' preferences. Furthermore, this model category accounted for the overall weak behavioral adaptation effects and the persistence of alignment observed when advisees preferred misaligned advice.

Parameter estimation was conducted by the *Stan* package [39] in R (Version 4.0.4) within the hierarchical Bayesian framework [40,41]. The candidate models were fitted to the data of all included participants (see S3 Text for details on model estimation, model selection). Model comparisons were performed by the *loo* package [42] to compute leave-one-out information criteria (LOOIC) for each candidate model. This approach allowed for a robust evaluation of model fit and facilitated comparison between different models [42]. We further verified our winning model using two rigorous validation approaches. First, we conducted a parameter recovery analysis for the winning model to assure that all parameters could be accurately identified and recovered (S3 Text and Fig A in S3 Text). Second, as model comparison provided relative performance of the winning model, we further performed posterior predictive checks for the wining model (S3 Text and Fig B in S3 Text), and we found that the posterior predictions captured key behavioral patterns at the trial-wise, individual, and

**Table 1. Candidate computational models and model comparisons.**

| Category | Model | # Par. | What influences $P$(align)? | ΔLOOIC (Study 3) | Weight (Study 3) | ΔLOOIC (Study 4) | Weight (Study 4) |
|---|---|---|---|---|---|---|---|
| Baseline | M1 | 1 | / | 974.3 | 0 | 648.6 | 0 |
| Social bias | M2 | 2 | Constant bias | 17.7 | 0.07 | 7.3 | 0.062 |
| | M3 | 2 | Advisees' confidence | 67.4 | 0.02 | 54.5 | 0.004 |
| | M4 | 3 | Acceptance | 216.6 | 0 | 112.8 | 0 |
| Social learning | M5 | 3 | Rescorla Wagner RL | 11 | 0.06 | 0.6 | 0.417 |
| | M6 | 4 | $a_{advice\ type}$ | 16.4 | 0.01 | 10.1 | 0.018 |
| | **M7** | **6** | $a_{advice\ type}^{feedback}$ | **0** | **0.85** | **0** | **0.499** |

*Note.* # Par., number of free parameters at the individual level. $P$(align), the likelihood of providing aligned advice. RL, reinforcement learning. $a_{advice\ type}$, learning rates differentiated by advice type (aligned/misaligned with advisees' opinion). $a_{advice\ type}^{feedback}$, learning rates differentiated by advice type (aligned/ misaligned with advisees' opinion) and feedback (acceptance/ rejection). ΔLOOIC, leave-one-out information criterion relative to the best-fitting model (lower LOOIC indicates better out-of-sample predictive accuracy). Weight, model weight computed with Bayesian model averaging using Bayesian bootstrap (higher model weight indicates relative higher probability of the candidate model to have generated the observed data). M7 (in bold) is the best-fitting model across Studies 3 and 4.

grand-average levels. Additionally, we conducted a full model recovery analysis to all the candidate models to ensure they were capable of the assumed behavioral patterns (S3 Text and Fig C in S3 Text).

*Baseline category (M1):* In M1, we assumed that advisors generated judgements for advice solely from their independent opinions (measured by their non-social advice in Session 1), and the independent opinions was accounted for by the value $Q_t^{ns}$, a two-element vector of the judgement options, specifying the value of the options 'higher' and 'lower' in trial $t$:

$$Q_t^{ns} = \left[ Q_t^{ns}(higher), Q_t^{ns}(lower) \right] \tag{1}$$

The non-social value of the selected option in the non-social scenario was fixed as 1, whereas the value of unselected option remained 0. Notably, this value of the selected option in the non-social scenario was scaled by a confidence index $\left| OS_t^{ns} \right|$ (measured by the wager on the selected option), such that higher confidence in the initially selected option corresponded to a greater tendency to advise the same judgement option again.

$$Q_t^{ns}(selected) = 1 * \Phi \left( \left| OS_t^{ns} \right| \right)$$
$$Q_t^{ns}(unselected) = 0 \tag{2}$$

where $\Phi$ was a normalization function ($\Phi(x) = \frac{x}{60}$) scaling the magnitude of opinions strength from the wagering scale (i.e., 1–60) to 0–1.

The probability ($p_t^{ns}$) of choosing an option was calculated using SoftMax function:

$$p_t^{ns}(higher) = \frac{1}{\left[ 1 + e^{-\tau \left( Q_t^{ns}(higher) - Q_t^{ns}(lower) \right)} \right]} \tag{3}$$

Finally, the advice judgement ($C_t$) was modeled by the categorical distribution:

$$C_t \sim categorical \left( p_t^{ns} \right) \tag{4}$$

*Social bias category (M2-M4):* This model category assumed that advisors considered their own independent opinions, while inclined to pursue congruence with advisee's opinions. In the models, advisors considered both the non-social value and the social value of each advice type (i.e., aligned/misaligned with advisees' opinions).

$$Q_t^s = \left[ Q_t^s(align), Q_t^s(misalign) \right] \tag{5}$$

M2 hypothesized that advisors conformed to advisees' opinions at a constant level (noted that, in all the subsequent models, in the trials that advisees' opinions were not presented, the model assumptions were identical to M1), captured by a consistent higher value of giving aligned advice compared to misaligned advice. To match the 0–1 scale of the non-social values, the social values in trial $t$ was set as below:

$$Q_t^s(align) = 1$$
$$Q_t^s(misalign) = 0 \tag{6}$$

M3 aimed to further address the possibility that advisors incorporate advisees' opinions to revise advice using their confidence as an indicator of credibility. This model hypothesized that the tendency of aligning with advisee's judgements increased with the magnitude of advisees' opinion strength ($|OS_t^{as}|$, measured by the investment magnitude) in their judgements:

$$Q_t^s(align) = \Phi \left( \left| OS_t^{as} \right| \right) \tag{7}$$

M4 was built on M2, assuming a pre-existing bias to give aligned advice to maintain interpersonal coherence. Moreover, this model further incorporated a reciprocal perspective of social influence [34], positing that individuals become more willing to align towards others' opinions when others were more acceptive of their own, and vice versa. Hence, M4 hypothesized that advisors' tendency of aligning could be increased after advisees' acceptance and decreased after rejection:

$$Q_{t+1}^{s}(align) = Q_{t+1}^{s}(align) + \kappa * R_t \tag{8}$$

where $\kappa$ (0 < $\kappa$ < 1) denoted advisor's relative sensitivity to the feedback $R_t$ (acceptance = 1; rejection = 0).

Across M2 to M4, advisors determined their judgements based on the integration of non-social and social values:

$$Q_t = w_{ns} * Q_t^{s} + w_s * Q_t^{ns} \tag{9}$$

where $w_{ns}$ and $w_s$ (0 < $w$ < 10) denoted advisor's cognitive weight assigned to non-social value and social value in the progress of decision, respectively.

Due to $Q_t$ was the value-vector of option 'higher' and 'lower', we mapped the $Q_t^{s}$ into these values based on the advisee's judgement in trial $t$ ($C_t^{as}$), to enable the combination:

$$Q_t^{s} = \begin{cases} [Q_t^{s}(align), Q_t^{s}(misalign)], & if: \ C_t^{as} = higher \\ [Q_t^{s}(misalign), Q_t^{s}(align)], & if: \ C_t^{as} = lower \end{cases} \tag{10}$$

Note that, in model specifications of the choice probability ($p_t$), we did not include the inverse Softmax temperature parameter as in M1 ($\tau$, in equation (3)). This was due to the non-social value and the social value of an option were explicitly constructed in a design-matrix fashion. Therefore, including the inverse Softmax temperature parameter would give rise to a multiplication term causing unidentifiable parameter estimation [41]. The probability ($p_t$) of choosing an option was calculated using an inverse logit linking function:

$$p_t(higher) = \frac{1}{\left[1+e^{-\left(Q_t(higher)-Q_t(lower)\right)}\right]} \tag{11}$$

*Social learning category (M5-M7):* This model category assumed that individuals engaged in value-based learning for each advice type (aligned vs. misaligned), with acceptance serving as rewarding feedback and rejection as punishing feedback, which trial-by-trial updated individuals' advice-giving tendencies before receiving feedback. Therefore, we described this dynamic using reinforcement learning (RL) models [23,40,43,44], a powerful framework to describe this learning process. This model speculates that individuals incorporate prediction errors (PE; difference between expected value and actual outcome of an action [44]) to update action value, and posits that individuals tend to choose the option with relatively higher expected value. Noted that, this category was designed to selectively examine whether advice-giving tendencies could be updated by advisees' feedback. Accordingly, for the initial values of $Q_t^{s}$, we set $Q_{t=0}^{s}(align)$ =1 and $Q_{t=0}^{s}(misalign)$ = 0, as in M2, to describe individuals' tendencies to give aligned advice before receiving any feedback, since M2 outperformed across M1-M4.

In M5, the updating of social value $Q_t^{s}$ was defined according to the classical Rescorla-Wagner model [44]:

$$Q_{t+1}^{s}(AT_t) = Q_t^{s}(AT_t) + \alpha_t * PE_t \\ PE_t = R_t - Q_t^{s}(AT_t) \tag{12}$$

where $PE_t$ denotes the prediction error between the feedback $R_t$ and the $Q_t^{s}$ of the chosen advice type $AT_t$ (aligned vs. misaligned), while the $Q_t^{s}$ of unchosen advice type remained unchanged. $\alpha$ (0 < $\alpha$ < 1) denoted the learning rate that

captured the weight of $PE_t$ in value updating. Specifically, the weak behavioral adaptation and the persistence of alignment observed when advisees preferred misaligned advice could be attributed to a small learning rate, which constrained adjustment from the pre-existing alignment bias.

M6 is constructed to serve as a contrast to M7, to test if the persistence of alignment could be defined in a simpler way. Specifically, this persistence could be understood as the overweight on the pre-existing bias ($Q_{t=0}^s(align)$) compared to the subsequent outcomes of giving aligned advice. This could be indicated by a smaller learning rate. Thus, we differentiated the learning rate of aligned and misaligned advice:

$$\alpha_t = \begin{cases} a_{al}, & if : AT_t = align \\ a_{mis}, & if : AT_t = misalign \end{cases}$$

(13)

M7 hypothesized that the persistence of alignment could be attributed to advisors' tendency to reinforce a belief that aligned advice leads to acceptance. In this case, a larger learning rate on acceptance and a smaller learning rate on rejection when giving aligned advice (compared to misaligned advice) could be expected. Therefore, we differentiated the learning rate of each association of advice type × feedback as follows:

$$\alpha_t = \begin{cases} a_{al}^{ac}, & if : AT_t = align, & R_t = acceptance \\ a_{al}^{re}, & if : AT_t = align, & R_t = rejection \\ a_{mis}^{ac}, & if : AT_t = misalign, & R_t = acceptance \\ a_{mis}^{re}, & if : AT_t = misalign, & R_t = rejection \end{cases}$$

(14)

### 4.2 Results

**4.2.1 Model-free analyses.** To illustrate advisors' behavioral adaptations to advisees' preferences, we first compared the likelihood of advisors providing aligned advice across different phases, where advisees demonstrated varying preferences, on two preference presenting orders. However, we primarily observed trends in these inter-phase advice-giving adaptations (all corrected $p$-values > 0.02, **Fig 3b**). To further explore results that could be hidden from grouping, we collapsed the ordering conditions and focused on the phases where advisees exhibited specific preferences for aligned and misaligned advice. The results showed that, advisors were significantly less inclined to provide aligned advice in the phase where advisees had preference for misaligned advice, as opposed to the phase where they had preference for aligned advice ($\beta$ = -0.19, $z$ = -2.69, 95% CI = [-0.32, -0.05], $p$ = 0.007) (**Fig 3b**). Descriptively, 60.4% (67 out of 111) of participants exhibited this direction of adaptation. Additionally, we also observed a negative relationship between judgement alignment and task progress (i.e., trial) ($\beta$ = -0.004, $z$ = -1.97, 95% CI = [-0.007, -0.00002], $p$ = 0.049), suggesting that advisors' pre-existing expectations about advisees' preference for aligned advice were dynamically updated with increased exposure to feedback.

**4.2.2 Model-based analyses.** To investigate the mechanisms behind advisors' adaptations and explore the potential reasons for advisors' persistence in providing aligned advice, computational models were constructed. The 'social bias' category assumed that, alignment bias could stem from intentions to preserve coherence or avoid advising responsibility, or from unintentional processes such as anchoring effects and opinion contagion. Consequently, advisors' propensity to provide aligned advice was rarely influenced by advisees' preferences. The 'social learning' category posited that alignment bias could be a learning-based phenomenon shaped by experience of repeated receiving feedback. From this perspective, advisees' preferences for aligned or misaligned advice, which were communicated through feedback as reward or punishment signals, could drive corresponding adaptations in advice-giving behavior to better match these preferences.

The results of model comparisons (**Table 1**) showed that, the models in the 'social learning' category (M5-M7) outcompeted the candidate models in other categories ('social bias' category, M2-M4) and the baseline model (M1). Within the 'social learning' category, M7 (**Fig 4**) outperformed the other candidate models, confirming that advisors exhibited learning biases during the feedback learning process. These biases were characterized by asymmetries in the learning rates, where the learning rate for feedback varied depending on whether advisors provided aligned advice and whether it was accepted.

Posterior predictive checks [43] on the winning model were conducted to test the congruence between true behavioral data and the simulated data at different scales: trial-wise scale, individual-wise scale, and grand average scale across trials and individuals (Fig B in S3 Text). Furthermore, the validity of the winning model M7 was further confirmed by linking computational parameters with the associated behavioral measurements. Specifically, parameter $w_s$ and $w_{ns}$ (in equation (9)) characterized the extent to which advisors guided their advice giving by the social value (i.e., the tendency of alignment) and by the non-social value (i.e., the certainty in their initial opinions), demonstrating a positive correlation for $w_s$ and a negative correlation for $w_{ns}$ with the overall judgement switch (from Session 1 to Session 2) (**Table 2**).

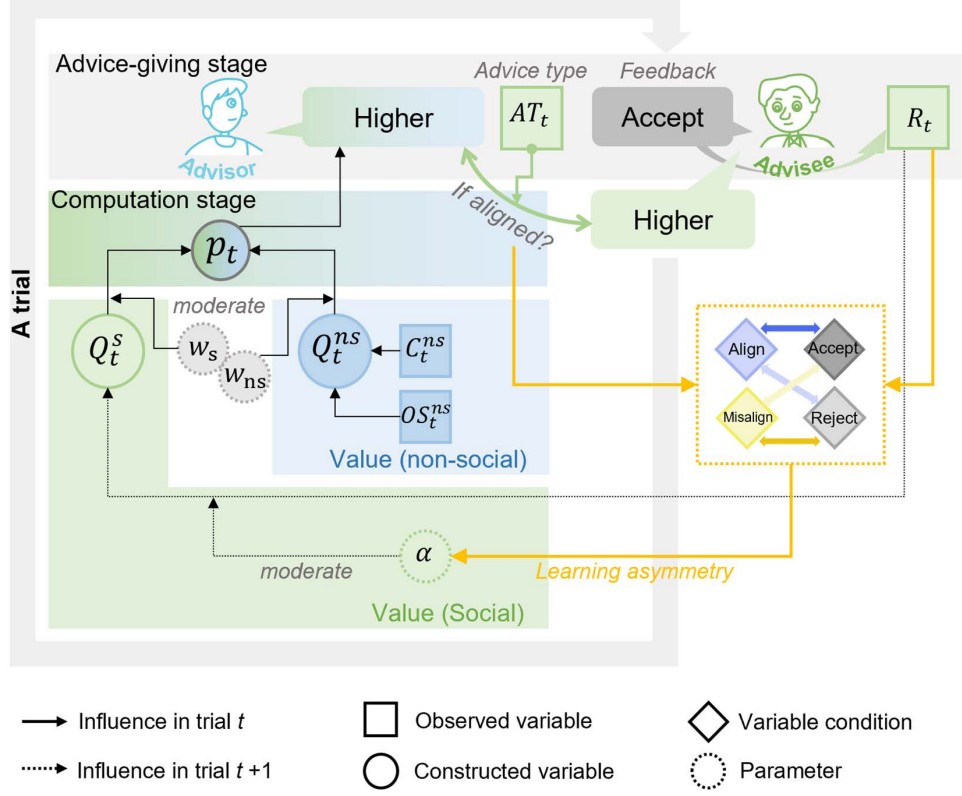

**Fig 4. The conceptual schematic of the winning model M7.** This conceptual schematic illustrates the psychological computation of advice-giving behavior within a single trial of interaction. Two parallel components (non-social value $Q_t^{ns}$, and social value $Q_t^s$) jointly determined the likelihood of choosing either judgement option ('higher' vs. 'lower').The non-social value $Q_t^{ns}$ reflected advisors' tendency to adhere to their personal opinion, as measured by their judgement option in the non-social session ($C_t^{ns}$), with the strength of this tendency indexed by their confidence ($OS_t^{ns}$), as scaled by post-judgement wagering magnitude. The social value $Q_t^s$ captured advisors' tendency to align or misaligned with advisees' opinions. The value of the chosen type (aligned vs. misalign) each trial was updated by the feedback from advisees ($R_t$) with acceptance serving as reward and rejection serving as punishment, with the updating pace scaled by the learning rate ($\alpha$). Notably, we modeled a confirmation bias in feedback learning (highlighted in yellow). Learning rates were parameterized separately for each combination of advice type and feedback type, allowing the model to capture asymmetries that reinforce the link between aligned advice and acceptance, and between misaligned advice and rejection.

**Table 2. Linking computational parameters with behavioral measurements.**

| $w_s$ ~ Overall judgement switch | | | | | |
|---|---|---|---|---|---|
| | **Study 3** | | | **Study 4** | |
| **r (s.e.)** | **df** | **P** | **r (s.e.)** | **df** | **P** |
| 0.36 (0.08) | 109 | < 0.001 *** | 0.24 (0.06) | 98 | 0.02 * |
| $w_{ns}$ ~ Overall judgement switch | | | | | |
| | Study 3 | | | Study 4 | |
| **r (s.e.)** | **df** | **P** | **r (s.e.)** | **df** | **P** |
| -0.68 (0.05) | 109 | < 0.001 *** | -0.65 (0.11) | 98 | < 0.001 *** |

*Note.* ***$P < 0.001$.

Next, we examined whether, and in what way, learning biases—captured by asymmetric learning rates across combinations of advice type and feedback type—hindered effective feedback learning, contributing to the overall propensity for providing aligned advice. As predicted, biases in advisors' feedback learning were observed. The learning rate of acceptance from giving aligned advice ($a_{al}^{ac}$) was larger than that from giving misaligned advice ($a_{mis}^{ac}$) (**Table 3**). Conversely, the learning rate of rejection from giving aligned advice ($a_{al}^{re}$) was smaller than that from giving misaligned advice ($a_{mis}^{re}$) (**Table 3**). Consequently, learning biases ($\Delta a$) could be gauged by the amalgamation of learning rate disparities in acceptance ($a_{al}^{ac} - a_{mis}^{ac}$) and rejection ($a_{mis}^{re} - a_{al}^{re}$), thereby yielding $\Delta a = \left( a_{al}^{ac} - a_{mis}^{ac} \right) + \left( a_{mis}^{re} - a_{al}^{re} \right)$.

Building upon the existence of learning biases, we validated the relationship between $\Delta a$ and participants' overall likelihood of providing aligned advice across Session 2 (**Table 4**). Furthermore, we also observed a negative relationship between $\Delta a$ and advisors' adaptations to advisees' preference for misaligned advice (**Table 4**), quantified by the shifts in the likelihood of giving aligned advice, from the initial phase to the phase where advisees preferred misaligned advice. On the contrary, we observed a positive relationship between $\Delta a$ and advisors' adaptations to advisees' preference for aligned advice (**Table 4**). These observations reinforced the notion that advisors' persistence in providing aligned advice was due to confirmation bias in feedback learning: advisors with larger learning biases found it more difficult to adapt to advisees' preference for misaligned advice, while being more readily able to adapt to advisees' preference for aligned advice.

**Table 3. The learning biases in feedback learning of advisees' preferences ($\Delta a$).**

| Study 3 | | | |
|---|---|---|---|
| $a_{al}^{ac}$ - $a_{mis}^{ac}$ | | $a_{al}^{re}$ - $a_{mis}^{re}$ | |
| **MD (s.e.)** | **P** | **MD (s.e.)** | **P** |
| 0.51 (0.01) | < 0.001*** | -0.32 (0.003) | < 0.001 *** |
| Study 4 | | | |
| $a_{al}^{ac}$ - $a_{mis}^{ac}$ | | $a_{al}^{re}$ - $a_{mis}^{re}$ | |
| **MD (s.e.)** | **P** | **MD (s.e.)** | **P** |
| 0.55 (0.01) | < 0.001 *** | -0.45 (0.01) | < 0.001 *** |

*Note.* ***$P < 0.001$. Parameter $a_{al}^{ac}$ and $a_{mis}^{ac}$ denoted the learning rates of acceptance from giving aligned advice and misaligned advice, respectively. Parameter $a_{al}^{re}$ and $a_{mis}^{re}$ denoted the learning rates of rejection from giving aligned advice and misaligned advice, respectively.

**Table 4.  Linking learning biases in feedback learning ($\Delta a$) with behavioral measurements.**

| $\Delta a$ ~ The overall propensity of providing aligned advice | | | | | |
|---|---|---|---|---|---|
| **Study 3** | | | **Study 4** | | |
| ***r* (s.e.)** | *df* | ***P*** | ***r* (s.e.)** | *df* | ***P*** |
| 0.73 (0.05) | 109 | < 0.001*** | 0.84 (0.03) | 98 | < 0.001*** |
| $\Delta a$ ~ Advice-giving adaptations to advisees' preference for misaligned advice | | | | | |
| ***r* (s.e.)** | *df* | ***P*** | ***r* (s.e.)** | *df* | ***P*** |
| -0.51 (0.07) | 109 | < 0.001*** | -0.35 (0.09) | 98 | < 0.001*** |
| $\Delta a$ ~ Advice-giving adaptations to advisees' preference for aligned advice | | | | | |
| 0.43 (0.08) | 109 | < 0.001*** | 0.23 (0.10) | 98 | 0.02* |

*Note*. *$P < 0.05$; ***$P < 0.001$.

## 4.3  Discussion

Despite moderate evidence from conventional behavioral analyses indicating advisors' adaptations to advisees' varying preferences, we found that advisors did show trial-and-error adaptations to these preferences through computational modeling, which was better able to capture the complex dynamics of advice interactions. Specifically, participants' behaviors in response to advisees' feedback were best explained by a reinforcement learning model, demonstrating that advice-giving tendencies can be gradually shaped by the feedback (acceptance/rejection) from advisees. Furthermore, model-based results suggested that advisors exhibited a confirmation bias in their feedback-driven adaptation process. Specifically, they tended to overweigh acceptance and downplay rejection when providing aligned advice. This bias provided a comprehensive explanation for the observed behavioral effects: advisors adapted to advisees' preferences while exhibiting a persistence to give aligned advice.

## 5.  Study 4

In Study 4, we further explored whether the behavioral adaptations in response to advisees' feedback persisted even when advisees were less receptive to advice, as indicated by their rare acceptance. This setup also allowed us to investigate whether alignment bias could be driven by a simple intention to evade advising responsibility, by comparing the extent of alignment bias exhibited by advisors in Study 4 to that in Study 3. If this intention was present, we would expect to observe a greater degree of alignment bias in Study 3, where advice was more frequently accepted and advisors thus bore more responsibility for the advisees' final decisions.

### 5.1  Methods

**5.1.1  Participants.**  We determined a sample size ($n = 112$) that resembled that of Study 3. Consistent with Study 3, participants who failed attention checks ($n = 7$) or demonstrated misunderstandings of the task in the post-task questionnaire regarding their belief of playing the role of advisors were excluded ($n = 5$). No participants expressed suspicion regarding the authenticity of the advisee in the post-task survey, and thus no exclusion was made on this basis. Consequently, 100 participants (mean age: 20.49 years, range: 18–27 years; 71 females) were included in the subsequent analysis.

**5.1.2  Experimental task.**  The experiment settings were mainly consistent with Study 3 (**Fig 3a**), except that insusceptible advisees were simulated in Session 2, who displayed a high resistance to taking advice, rejecting advice in most of the time (overall acceptance rate = 30%, the acceptance rate for each type of advice—aligned or misaligned with advisee's judgement—was distributed across three levels: 10%, 30%, and 50%, **Fig 3c**). For example, as advisees exhibited phased preference for aligned advice, aligned advice would be accepted at a higher rate (50%) than misaligned advice (10%), and vice versa.

**5.1.3 Design, measurements, and analytical plans.** We employed identical designs and analytical plans in accordance with Study 3. Additionally, to further examine whether alignment bias was driven by an intention to evade advising responsibility, we first implemented a post-hoc manipulation check, examining whether participants in Study 3, where advisees accepted advice as their final decisions to an overall greater extent, experienced a stronger sense of advising responsibility compared to those in Study 4. Specifically, we first compared post-survey ratings of the item "*I made my advice after careful considerations*" between participants in Study 3 and Study 4. Behaviorally, we further confirmed this notion by investigating whether participants in Study 3 advised more cautiously, as reflected by a greater reduction in investment magnitude in Session 2 relative to Session 1 (baseline level). Based on this manipulation check, we constructed a linear mixed-effect model to compare the extent to which advisors aligned their judgements with those of advisees in Study 4 to Study 3, aiming to rule out the possibility that the alignment bias was merely driven by a desire to evade advising responsibility—an alternative account that predicts greater alignment in Study 3.

Furthermore, a linear mixed-effect model was constructed to compare task engagement between participants in Study 4 and those in Study 3, by examining the frequency of attention check failures. This analysis aimed to rule out the possibility that the absence of significant behavioral adjustments observed in Study 4 was simply driven by reduced engagement or motivation.

## 5.2 Results

In study 4, we only observed trends in the inter-phase variations of advisors' tendency to provide aligned advice (all corrected *p*-values > 0.29, **Fig 3c**). After collapsing the ordering conditions, we still failed to find a significant reduction in advisors' inclination to provide advice aligned with advisees' opinions in the phase where advisees preferred misaligned advice, compared to the phase where advisees preferred aligned advice ($\beta$ = -0.10, $z$ = -1.42, 95% CI = [-0.04, 0.25], $p$ = 0.16). Descriptively, only 42% (42 out of 100) of participants exhibited this directional adaptation. Additionally, a non-significant negative relationship between advisors' judgement alignment and the acquisition of feedback (measured by the number of trial) was observed ($\beta$ = -0.001, $z$ = -0.64, 95% CI = [-0.005, 0.003], $p$ = 0.52).

Although we did not find evidence for advice-giving adaptations at the behavioral level, the results from model-based analyses confirmed that the adaptation to the feedback from advisees persisted. The results of model comparison showed that, M7 outperformed all other candidate models (**Table 1**) and accurately captured the true behavioral data in Study 4 (Fig B in **S3 Text** and **Table 2**).

Furthermore, the relationship between learning biases ($\Delta a$) (**Table 3**) and advisors' propensity to provide aligned advice were mostly observed, except for a non-significant result in the relationship between $\Delta a$ and advice-giving adaptations to advisees' preference for aligned advice (**Table 4**). Notably, larger learning biases ($\Delta a$) were observed in Study 4 compared to Study 3 (Mean Difference = 0.18, $t$(209) = 10.85, 95% CI = [0.15, 0.21], $p$ < 0.001), suggesting that advisors experienced more hindered feedback learning. Additionally, participants in Study 4 exhibited fewer attention check failures compared to those in Study 3 ($\beta$ = -0.26, $z$ = -2.39, 95% CI = [-0.47, -0.05], $p$ = 0.02), suggesting that the intensified learning bias and the lack of significant behavioral adjustments observed in Study 4 were unlikely to result from reduced task engagement.

To reinforce the notion that advisors' alignment was not simply driven by a desire to avoid responsibility evidenced by the relatively weaker performance of M2–M4, we further compared participants' behavioral patterns between Study 3 and Study 4. Compared to Study 4, advisors in Study 3 demonstrated a greater sense of responsibility in giving advice provided more conservative advice, as reflected in higher post-survey ratings of the item "*I made my advice after careful considerations*" (Mean Difference = 0.27, t(209)=2.37, 95% CI = [0.05, 0.50], $p$ = 0.02), as well as a greater tendency to provide advice cautiously (measured by a greater reduction in investment relative to their baseline level) ($\beta$ = 1.42, $t$(27394.1) = 4.46, 95% CI = [0.80, 2.05], $p$ < 0.001). Based on this evidence, we next asked whether advisors who experienced a stronger sense of responsibility were more likely to align with advisees' opinions as a means of evading this responsibility.

The results showed that, participants did not exhibit a greater degree of alignment with advisees' judgements in Study 3, where they were more responsible for advisees' final judgements, compared to Study 4 ($\beta = 0.03$, $z = 0.83$, 95% CI = [-0.09, 0.04], $p = 0.41$).

### 5.3 Discussion

In Study 4, we replicated most of the findings from Study 3. Although model-free analyses did not reveal evidence of behavioral adaptations to advisees' feedback, model-based analyses demonstrated that advisors' behavioral adaptations induced by advisees' feedback persisted. Consistent with Study 3, participants' adaptations to advisees' preferences were best captured by a biased learning model. Notably, larger learning biases were observed compared to Study 3, echoing with the absence of behavioral evidence indicating advising adaptations.

Additionally, by further comparing the level of alignment with advisees' opinions exhibited by advisors when they were more responsible (Study 3) versus less responsible (Study 4) for advisees' final decisions, we provided further evidence against the notion that alignment bias was driven by a constant intention to evade advising responsibility.

## 6. General discussion

Advice interaction is commonly perceived as a uni-direction transmission of social influence from advisors to advisees [1,3]. The bi-directional information exchanges within this interactive process, particularly the transmission from advisees to advisors, remains largely neglected. We addressed this gap by unveiling how and why advice giving is influenced by advisees' opinions. The alignment bias, characterizing advisors' inclination to conform to advisees' initial opinions, may be gradually shaped by feedback from advisees via a value-based learning mechanism in everyday interactions (see **Fig 5** for a graphical summary).

In our work, we simulated a real-time interaction context that included social consequences (advisees' acceptance or rejection of advice) to examine whether advisors naturally tend to align their advice with advisees' opinions when given access to them. Our findings demonstrated that, advisors have a robust propensity to align their advice with the opinions of advisees, whether in an evaluation-based incentive context, embedded in everyday advice interactions, or in a performance-based incentive context, where advice accuracy was explicitly encouraged. Deciphering this phenomenon, we found that advisors consistently re-aligned with advisees' incongruent opinions, even when those opinions were incorrect, revealing a tendency for opinion conformity. Crucially, our data showed that this conformity could not be rather attributed to over-reliance on irrelevant information of judgement accuracy or low-level processes (e.g., information seeking, response copying, or unconscious susceptibility to others' opinions). Instead, our data demonstrated the normative nature of this conformity. This was evidenced in a post-interaction session, where re-alignment with advisees' opinions was maintained to a reduced degree once their interaction ended, compared to when participants were told they would continue interacting with the same advisee. Importantly, this difference could not be explained by a tendency to maintain self-consistency (by repeating previous judgements) in the presence of the same advisee.

With the introduction of acceptance and rejection of advice provided by advisees, our computational modeling analyses revealed that advisors exhibited feedback-driven adaptations in their advising tendencies (to give either aligned or misaligned advice) with advisees' fluctuating preferences. These behavioral responses to feedback were best captured by a Reinforcement Learning model. This model was grounded in an evolutionary perspective, assuming that individuals' alignment bias may be shaped by everyday experience of advice-giving, whereby advisees exhibit a common preference for advice that aligned their initial opinions. Therefore, as advisees' specific preferences became evident through their feedback of advice, advice-giving tendencies—providing either aligned or misaligned advice—shifted in the direction of maximizing acceptance and minimizing rejection. Moreover, even when acceptance from advisees was scant, the feedback-driven adaptations persisted, as bolstered by the model-based evidence, further suggesting that advice-giving behavior is sensitive to its consequence.

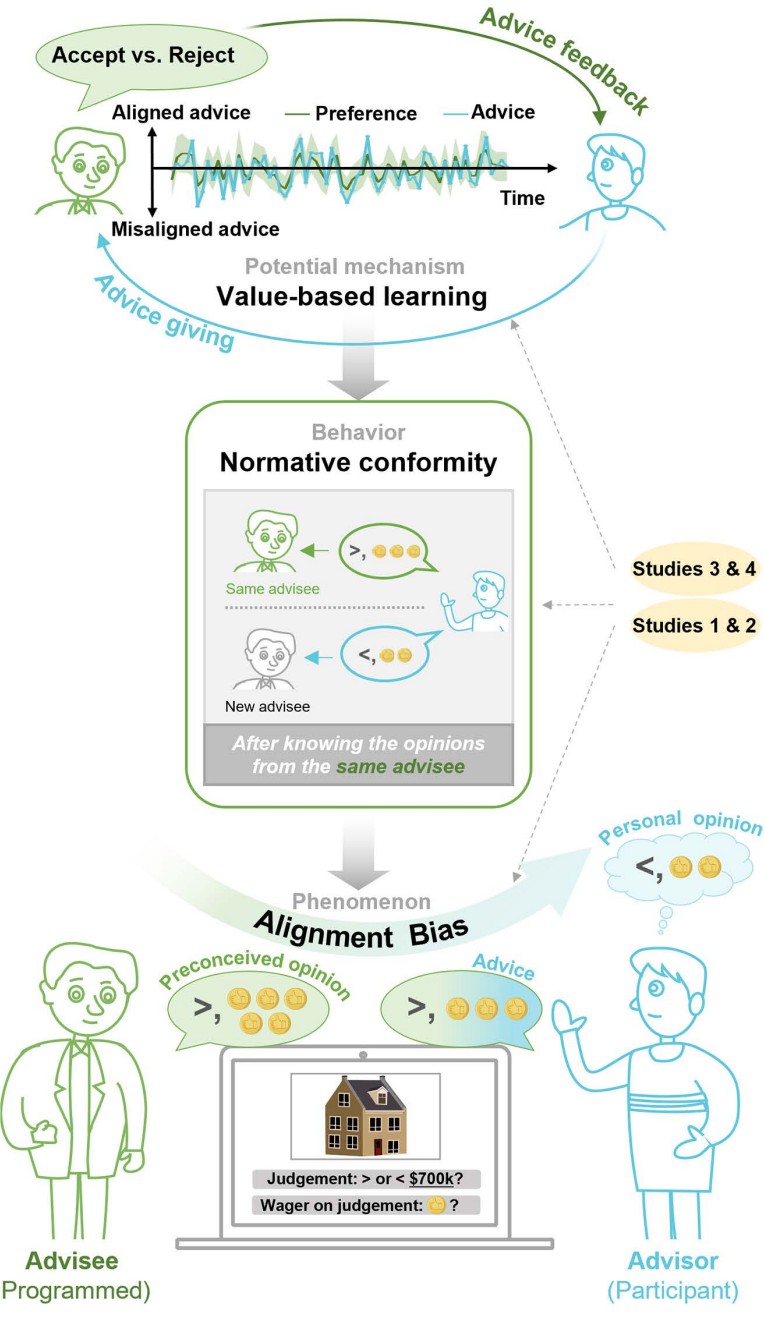

**Fig 5. A graphical summary of the key findings of the current study.** In Studies 1–2, we identified an alignment bias in social advice giving, whereby advisors adjusted their advice toward advisees' initial opinions. We further demonstrated that this bias was driven by normative conformity, as it disappeared once the interacting advisee changed. In Studies 3–4, we discovered advisors' behavioral adaptations that maximized advisees' acceptance and minimized rejection of advice, providing a value-based learning perspective on the emergence of alignment bias.

However, despite adaptations towards advisees' varying preferences, advisors persisted in a propensity to align with advisees' opinions throughout the interaction. Our model-based evidence showed that, advisors tended to overweigh acceptance and downweigh rejection following the provision of aligned advice, reflecting as learning biases in the advice

interaction. This led to inefficiencies in updating their pre-existing expectations about advisees' preferences for aligned advice. In other words, the ingrained belief of advisees' preference for aligned advice, can selectively amplify advisors' sensitivity to the evidence that supports the pre-existing association between aligned advice and acceptance, and dampen their sensitivity to the evidence that violates this belief. The observation of learning biases echoes with the phenomenon of confirmation bias [45,46], a ubiquitous cognitive bias which has been widely documented, indicating that individuals tend to disregard information contradicting their existing beliefs, which has also been observed in various interaction-based learning processes [47,48]. Intriguingly, these learning biases intensified when acceptance from advisees was rare. This observation aligned with the lack of behavioral evidence indicating that advisors adjusted their advice-giving tendencies under this condition. According to cognitive theories of stress [49], stress can impair adaptive outcomes such as social functioning and cognitive capabilities. Specifically, acute stress—a transient response to a recent stressor—is associated with reduced cognitive flexibility [50] and adaptive behaviors (such as instrumental learning) [51,52]. Moreover, experiencing rejection from others is associated with acute stress [53]. From this perspective, frequent rejection of advice by advisees may induce stress, thereby impairing advisors' flexibility in adapting to advisees' preferences and negatively affecting their learning outcomes.

Zooming in, our findings closely align with the research on advice giving, demonstrating advisors perceive acceptance from advisees as reward [4,6,26], and their dislike of being disregard by advisees [4,5]. Our studies suggest that advisors are learned-to-be experts on the behavioral tendencies of advisees. As shown in Hertz et al. (2017), advisors tend to use competitive strategies to secure acceptance, i.e., advising over-confidently when ignored by the advisees and diffidently when trusted by the client. This situation-dependent strategy resonates with previous research showing that advisees [54], as well as information receivers [35], place more trust in advice or information conveyed with higher confidence; however, as the advisors' confidence is found to be unindicative of accuracy, it is perceived as arrogance, and acceptance drops accordingly [36]. Our studies suggest a mechanistic account of why these two veins of findings are closely linked: these observed advising tendencies or biases could be a long-term consequence of social learning about advisees' behavioral heuristics or preferences.

A recent study provides further insight into the variability of individuals' willingness to offer opinions as advice [25]. Individuals tend to share their opinions as advice when they are more confident in their performance. However, individuals are also more likely to advise when informed that their advice would be presented to future participants with their performance ratings, indicating that advice-giving behavior is not only shaped by perceived performance but also by social incentives such as reputational rewards. In line with previous findings showing that individuals modulate their confidence to enhance social influence, these results further support the view that, beyond the internal motivation to offer accurate advice, advice-giving behavior is also underlain a desire to manage reputation. On the other hand, research on social exclusion offers another perspective: the act of securing acceptance may also stem from an aversion to rejection. Social rejection is inherently aversive, as it signals threats to one's sense of belonging and self-worth [55,56]. As demonstrated by research on normative conformity, the need to belong even often overrides the pursuit of accuracy [13,34,57]. Taken together, these findings—highlighting both the desire for social rewards and the avoidance of social punishment—provide a deeper understanding of the motivational basis underlying advice-giving behaviors, and further elucidate why social outcomes may ultimately introduce systematic bias and compromise the accuracy of advice.

Zooming out, our observation of advisors' evolutionary adaptations in navigating acceptance through feedback-driven learning well corresponds with the reward learning widespread in the social world. Shown by a large body of literature, individuals are adept at identifying the attributes mostly associated with reward within a specific context and engage in goal-directed learning on these attributes rapidly (i.e., model-based learning) [58,59]. Even when the generative structures of reward are latent, individuals can quickly infer which object is rewarding through trial-by-error model-free learning [23,24]. Consequently, these latent learning processes proficiently direct and optimize our behaviors within interactions where potential rewards are embedded. However, from a game-theoretic perspective, when advisees express their

opinions before seeking advice, the situation creates a functional dilemma that exposes an inherent conflict between advisors' social interests and the epistemic utility of advice for advisees. In other words, while advisors may pursue alignment to gain approval and satisfaction from advisees, such aligned advice may simultaneously lose epistemic value and be underappreciated. This tension underscores the need for future research to conceptualize advice-giving as a deliberative trade-off between epistemic accuracy and reputational or relational gains. Moreover, identifying when and how advisees recognize and adjust for biased alignment could offer valuable insight into the reciprocal dynamics that characterize multi-round social interactions.

This study has several limitations. First, we examined alignment bias in an estate property evaluation task, a domain unfamiliar to student participants. This was designed to simulate everyday advice-giving situations where advisors and advisees share similar levels of expertise, such as choosing a restaurant or making a purchasing decision. However, whether alignment bias persists among experts remains unclear, as professionals in fields like finance, medicine, or law may exhibit different patterns due to greater epistemic confidence. Second, we investigated whether advisors adapt their advice-giving behaviors in response to repeated feedback of advice, by manipulating it in a binary form (i.e., acceptance or rejection). While this approach ensured that advisees' preferences were easily discernible, real-world advice-taking is often more nuanced, with advisees integrating advice to varying degrees rather than making strictly binary decisions. Future research should adopt more ecologically valid paradigms, such as longitudinal studies tracking repeated interactions or experimental designs incorporating graded advice integration.

Moreover, although our design shows that advisors adjust their behavior in ways that tend to maximize acceptance and minimize rejection, it cannot determine whether this process reflects a conscious goal. Future studies could incorporate self-report-based assessment to assess whether participants deliberately attended to this feedback to guide subsequent adjustments in their advice. Finally, in the current study, we induced participants' perception of engaging in social interactions with a human partner through physical contact and task-based reward interdependence, to ensure that participants believed they were receiving advice feedback in real social encounters. Although this approach proved effective, the extent to which our findings on alignment bias generalize in real-world social relationships (e.g., friendships), where advice feedback may carry greater value for individuals, remained to be investigated. Addressing this issue is essential for clarifying the social complexity of alignment bias, and future research should examine the interplay between the prosocial motivation to benefit close others and having favorable social feedback during social interaction. Future studies could further probe the social complexity of alignment bias by comparing advisors' responses to human advisees versus non-social targets (e.g., AI), to examine whether the observed unselective re-alignment patterns differ between these contexts.

In summary, this study presents a comprehensive investigation into the alignment bias. Deciphering this phenomenon, we illustrate that advisors tend to conform to advisees' preconceived opinions, which is potentially shaped by a social learning mechanism where acceptance and rejections from advisees serving as drivers. These results shed new light on the reciprocal nature of social influence [34]—the individuals who influence and perceive successful influence as a form of reward may, therefore, be susceptible to the reactions of those whom they seek to influence—within the widespread informational interactions.

## Supporting information

**S1 Text. Pretests on the stimuli.**
(DOCX)

**S2 Text. Mixed-effect models specifications.**
(DOCX)

**S3 Text. Model estimation, model selection, parameter recovery and model recovery.**
(DOCX)

**S1 Fig. The frequency of opinion shift rate (OSR) among all trials of each participant in Study 2.**
(TIF)

## Acknowledgments

We thank Dr. Feng Sheng for providing valuable comments and suggestions on an earlier version of the manuscript. The computations described in this paper were partially performed using the Birmingham Environment for Academic Research (BEAR).

## Author contributions

**Conceptualization:** Xitong Luo, Lei Zhang, Yafeng Pan.

**Data curation:** Xitong Luo.

**Formal analysis:** Xitong Luo.

**Funding acquisition:** Yafeng Pan.

**Investigation:** Xitong Luo.

**Methodology:** Xitong Luo, Lei Zhang, Yafeng Pan.

**Supervision:** Yafeng Pan.

**Visualization:** Xitong Luo.

**Writing – original draft:** Xitong Luo.

**Writing – review & editing:** Lei Zhang, Yafeng Pan.

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
