## [Decision Letter · Decision Letter 0]

17 Jun 2025

Do we advise as one likes? The alignment bias in social advice giving

PLOS Computational Biology

Dear Dr. Pan,

Thank you for submitting your manuscript to PLOS Computational Biology. After careful consideration, we feel that it has merit but does not fully meet PLOS Computational Biology's publication criteria as it currently stands. Therefore, we invite you to submit a revised version of the manuscript that addresses the points raised during the review process.

Please submit your revised manuscript within 60 days Aug 17 2025 11:59PM. If you will need more time than this to complete your revisions, please reply to this message or contact the journal office at ploscompbiol@plos.org. Please include the following items when submitting your revised manuscript:

We look forward to receiving your revised manuscript.

Kind regards,

Zhiyi Chen

Academic Editor

PLOS Computational Biology

Tobias Bollenbach

Section Editor

PLOS Computational Biology

**Additional Editor Comments :**

Thank you for the patience. I have received all the review comments to this submission, largely varying from minor revision to rejection. In conjunction with my personal evaluation, I recommend a major revision for this manuscript, and ask authors to carefully address methodological and conceptual concerns that Reviewer #1 and 2 raised.

**Journal Requirements:**

At this stage, the following Authors/Authors require contributions: Xitong Luo. Please ensure that the full contributions of each author are acknowledged in the "Add/Edit/Remove Authors" section of our submission form.

3) We have noticed that you have cited [S3 Fig] on page [47]. However, there is no corresponding file uploaded to the submission. Please upload it as a separate file with the item type 'Supporting Information'.

Potential Copyright Issues:

i) Figures Graphical Abstract, 1, 2A, 3A, and 4B. Please confirm whether you drew the images / clip-art within the figure panels by hand. If you did not draw the images, please provide (a) a link to the source of the images or icons and their license / terms of use; or (b) written permission from the copyright holder to publish the images or icons under our CC BY 4.0 license. Alternatively, you may replace the images with open source alternatives. See these open source resources you may use to replace images / clip-art:

5) Thank you for stating "The authors declare no conflicts of interest associated with this manuscript." If you have no competing interests to declare, please state "The authors have declared that no competing interests exist".

**Reviewers' comments:**

Reviewer's Responses to Questions

Reviewer #1: The article offers an interesting account of four experiments seeking to understand the social mechanisms behind advice-giving behavior. The statistical analysis is well done and the series of experiments build on one another.

I have three areas of concern, in the following order:

First, the alignment of the research aims and the experimental design, second the interpretation of the results, and third the explanation of the series of events in the studies.

Alignment of aims and design

I appreciate the article weighing the distinctions among rejection, congruence, and inaccuracy as distinct factors in advice-giving. However, I am concerned that the real estate game as described does not align with a study of social advice per se, which the study describes as "informationally rich" and potentially influenced by "our need to connect with others". The study describes a low-information, low-sociality environment, with no social connection possible. Relationships by their nature involve multiple rounds of interaction, and advice giving in a interpersonal connection setting calls upon this relational history and mutual understanding. It is unclear how these traits of social advice and relationships map into a game wherein an individual is seeking to guess a numerical value and express confidence in their judgment, with and without an additional piece of information. It seems more likely to test a version of economic decision-making.

I think I have read correctly that the tasks are performed with a few iterations at most and with no tracking in the interface showing performance and alignment that an advisor could then be understood to be able to analyze and learn from; this kind of design would address some of my concerns (e.g., it makes learning and rational adaptation more possible) but not all of my concerns given the lack of richness, lack of information / context, and lack of sociality that would persist.

Interpretation of results

In study 1, the article states: "Participants were informed that their payments were partially determined by the advisees given their subjective evaluations on the advice at the end of the task." It seems reasonable for participants to expect to be penalized by the subjective evaluation for misalignment if they are not correct, and rewarded for alignment regardless of whether they were correct or not, making misalignment a higher-risk state. I believe M7 seeks to evaluate this interpretation, however this distinction and the problem it poses for study 1 is not addressed.

The article describes its aim as "to investigate whether advisors gradually aligned their advised investments to align with advisees’ risk preference". However, the description of the stimuli does not incorporate discussion of risk preference, only the dollar figure guess. In addition, it is unclear how the study tests notions of gradual alignment in a two-round interaction.

Study 2 reiterates the interpretation from Study 1 that tendency to align with incongruent opinions as "alignment bias" and proposes to distinguish this from two other potential interpretations -- opinion contagion and anchoring. This makes sense, however the design seems insufficient to eliminate contagion and anchoring as interpretations --- or as more simple information-seeking / rational choice behaviors as mechanisms given the low information conditions and few iterations.

The success of M7 and general persistence of desire for advice acceptance is interesting, but ultimately it is situated inside a series of studies that are structured to evaluate economic decisionmaking rather than social relationships.

Clarity of design overall

The essence of my concern here is that it is not clear what participants know and when they know it.

For example, for study 1 the article describes realignment as occurring "even when incorrect" and evaluating whether advisor realignment was "contingent on the accuracy of those opinions" -- however, it is not clear how and when the participants knew the actual value, or how the magnitude of the incentive for making a good guess compared to the incentive of the advisee giving them a positive rating.

The same issue persists for study 2, which describes how "the inclination of re-align persisted both when advisee’s judgement was correct...and when it was incorrect." Did advisors re-align to match the opinions of advisees despite knowing that the advisee was incorrect and that the advisee would have no influence on their reward? The incentive structure in the design of study 2 further diverges from the incentive structure of more naturalistic advice-giving scenarios, which socially reward advice acceptance and accuracy in varying degrees by context with the advisee as a substantial factor. In general the reward conditions and subjects' understanding of the reward conditions needs to be made very clear for each study.

I further noted the usage of random variation from the actual pricing of real estate as part of the experimental manipulation; did the algorithmically-generated figures conform to common expectations around what a price looks like (e.g. round numbers, common digits), or did the numbers potentially appear artificial? I also observe that the study filtered out people who detected the manipulation: were people who answered this question informed that their reward would not be influenced by the answer to the question ? What was the magnitude of individuals who were excluded due to this? Were there systematic differences in the performance of these more insightful individuals with respect to the task?

Document preparation notes:

Line 119 -- "whether acceptance-seeking intentions underlying this bias."

Line 159 -- "a sample of at least 59 participants were required"

Line 500 -- "the inclination of re-align persisted"

Reviewer #2: I thank the authors for producing this manuscript, which constitutes a timely addition to an important and gravely understudied topic. I also thank the editor for the opportunity to review this work. I want to formally state as well that I have not used AI for any part of this review process. Here is a more in-detail account of my views on each part of the manuscript.

GRAPHICAL ABSTRACT

I believe it is very good, but it may be a bit opaque for people who are not initiated in the topic. The upper part (studies 3 and 4) is super clear, but I think the lower part (studies 1 and 2) is a bit less transparent for the untrained eye. I do not request any modifications because if I'm the public for this abstract, then it's spot on. But in case you may want a more accessible item, maybe consider rethinking the lower part.

INTRODUCTION

The introduction is principled and complete. The question of the relevance of advice-giving and the biases that intervene when crafting advice are well defined. The one big omission that I think should be addressed for clarity and completion is a mention to *how* advice-givers consider their *own* knowledge when deciding whether or not to give advice. This was studied extensively by Anllo, Hertz and cols (see https://www.nature.com/articles/s44271-024-00175-5). Incorporating this angle into the introduction or discussion is I believe of crucial importance, because that work outlies an adversarial setup in which participants have to weight their own knowledge about value versus different social cues at the time of giving advice (an issue that is complementary to the researchers' question to a very high degree, because it touches directly upon the key question of whether advice is useful or not). Another important point is that the idea that advice-gives could be influenced by the very people they are trying to influence (the advisees) is important and needs to be studied, but it is not new as it has been considered (although not directly) in the work by Hertz et al 2020, where advice-givers seek to sway advisees (although my remark here is less severe because the authors do come back to this paper during their Discussion).

STUDY 1

METHODS

-I appreciate the a-priori power analysis, but would like to see some justification as to why a d = 0.4 was chosen, and why a power of 85% was the target.

-I would appreciate a bit more clarity in the presentation of the design. If I understand correctly, even in the "social" condition participants are never in actual contact with the advisees, right? Further, you mention that you generate advisees judgements, but you also mentioned that you needed to keep the order of blocks as "isolation" then "social" because you needed the decisions and judgements of the advisees. So please exactly explain how much of the advisees judgemetns is being engineered or "generated". Is it just the order? Then the investments were also randomly selected from normal distributions... please attempt to explain exactly how much is "human made" and how much is "generated".

-When it comes to your ANOVA approach, to my knowledge the term "mixed-effects one-way ANOVA" does not exist. It appears you are conflating linera mixed-effect regressions and x-ways ANOVAS. Please clarify.

-You discuss several "mixed-effect" approaches, which implies you actually built a statistical model (an assumption you confirm two paragraphs later... it may be better to explain everything together in one go). However, no information is provided however as to what these models are (just presenting the models and the approach for parameter optimization would suffice). Did you do any testing for goodness of fit?

-Please be more transparent in your exclusion criteria, I understand you outlined them in S2 Text, but still, I do not consider it good practice to hint in the main text that additional exclusions took place without saying exactly what happened.

RESULTS/DISCUSSION

-Perhaps my only major query with these findings is that (unless I misunderstood) the incentive structure of advice-givers is mostly rooted on advisee's performance/rewards. Namely, it could be argued that participants focus on matching advisee behavior because they do not have an immediate consequence if they are wrong. I know the incentive structure is not described fully in detail so maybe I'm missing something here, but it feels like the advisees don't really face any kind of consequence if they decide to completely disregard their own opinion and just match the advisee's. This is important not only on a higher order, but also because as it stands the trials are build in a way that advisee's judgement could be priming advice-giver responses on a lower-level processing stage (i.e. advice-givers just repeating what they just saw on screen because it is less cognitive effort).

-You mention a negative relation between advisor's confidence in non-social judgements and and judgement realignment. Please explain this further, as this may very well be evidence of advice-givers not over-relaying on advicees confidence, but not like this. Namely, if I understand correctly (but correct me if I don't), the more advisers trusted their judgement the less they changed their advise to match advisee's. But if this is the case, then it becomes somewhat important to probe how much advisees and advice-givers know about the real estate market, or even their socio-economic status. The reason why I'm saying this is because, as it stands, it could be that the overarching rule advisees are implementing is "If I'm very confident about this one opinion I go with my own thoughts, but otherwise I default to advisees". Further testing (or at least, discussion) would be needed to rule out whether advice-givers are trying to re-align with advisees or if they are simply using them as an extra source of information to reduce their own uncertainty and realign their choices.

-

STUDY 2

-I understand the rationale of this study, but my worry from the previous study compounds here: you say that participants would not persist in their realignments when facing new advisees if they acted on intention. However, if advisee's judgement is being treated as a supplementary source of information and nothing else, then advisee's being known or unknown would make no difference (which is exactly what you find in your results). Other than that I really appreciate the fact that you changed your incentive here. Good call!

-Why use a post-task survey here and not in the previous study?

STUDY 3

METHODS

-The study design is clear and well-motivated, and I appreciate the addition of a pre-registered replication sample. However, the setup may not entirely support the authors' conclusion that misalignment avoidance is driven by intentional strategic behavior. Specifically, the manipulation involving advisee feedback being random or contingent is really good, but it assumes participants actively track and integrate this feedback which may not be the case. Some measure of whether participants noticed or used this contingency would greatly strengthen the argument. A post-task inference test could have addressed this issue (but I don't expect you to redo the experiment just for this!).

-The framing of the “intentionality” behind advice realignment feels somewhat overstated. It is not clearly demonstrated that participants understand the feedback as a signal for acceptance, nor that they adjust their behavior strategically in response to it. As in Study 1, it is still a distinct possibility that participants in the contingent condition act based on features of the feedback, and not necessarily because they are engaging in deeper intention-driven behavior.

-Again, similar to Study 1, you refer to “mixed-effects ANOVAs,” which seems terminologically imprecise. I suggest naming them as linear mixed-effect models, and explicitly stating the structure (fixed and random effects, distributional assumptions, etc.). Likewise, providing model fit metrics would help me (and the readers) appreciate the strength of the conclusions.

-Concerning the models, please keep in mind that you may be over-parametrizing, which could in turn lead to over-fitting. Since in your approach you sort of "bake" your hypothesis into the model, and you don't really test adversarial models, some consideration on why this approach is not circular would be greatly appreciated. Some predictive use of your models or parameter recovery could help drive the point across too (I'm not requesting that you do everything, but at least something other than LOOIC)

RESULTS/DISCUSSION

-The key finding—that advice-givers align more under contingent feedback—is indeed interesting, but it doesn’t necessarily imply that participants are intentionally modifying their advice to gain acceptance in a conclusive manner. The effect could also be driven by a more general reinforcement-like process (e.g., repeating previously accepted advice), without a goal-level intention to influence others. The authors should clarify whether they consider intention and reinforcement separable, and more importantly, whether they can be distinguished in this design. Overall some nuance would be appreciated.

-Finally, the authors do not report (in the main text) any individual differences analyses so far. Did all participants respond similarly to the contingent vs yoked feedback? If not, understanding who adapts and why could help uncover whether the behavior is strategic or reactive.

STUDY 4

METHODS

-Beware, reducing advisee's acceptance ratio to 30% may serve the purpose of your manipulation but it is also rendering feedback considerably scarcer. This may have implications on the results.

-Again, consider rewriting your mixed-ANOVA approach.

-Again, do not relegate exclusions to supplementary materials.

RESULTS/DISCUSSION

-I realize as I see that indeed the models confirm the idea that there is a learning bias, that no efforts have been made to ensure these results don't come from low motivation or task disengagement.

-The between-study comparison meant to test the “responsibility avoidance” hypothesis—i.e., that advisors might align more in Study 3 (higher acceptance) than in Study 4 (low acceptance)—yields null results, but this comparison is underdeveloped. It’s treated as a key test but presented in a single sentence. If responsibility avoidance were a genuine driver, we would expect more behavioral alignment when the risk of influencing others is higher. But the absence of such a pattern does not prove its irrelevance; it might instead suggest the manipulation of “responsibility” (via advisee acceptance rates) is not strong or explicit enough to affect advisors’ sense of accountability. A more nuanced or self-report-based assessment of responsibility perception might yield better insights.

-I worry again here that the weight of the theoretical implications of model fitting may be overstated. M7 still largely describe behavioral patterns rather than unambiguously infer psychological goals. The authors argue that persisting use of aligned advice despite evidence it won’t be accepted reflects biased feedback learning. This is plausible, but again, alternative interpretations (e.g., advice repetition as habit, default response, or even frustration-based disengagement) cannot be ruled out.

GENERAL DISCUSSION

It is overall principled and complete, relying adecquately on the findings of all previous studies. I do insist however that this would be an adecquate time to consider performance as an internal motivator to give advice, as discussed by Anllo, Hertz et al 2024 (see https://www.nature.com/articles/s44271-024-00175-5). I would also suggest that this general discussion also echoes some of the caveats and calls for caution that I mentioned above. I would encourage the authors to dial back the confidence of their conclusions or at least explicitly acknowledge these limitations.

Reviewer #3: This manuscript presents a novel and important perspective on "alignment bias" in advice-giving behavior, emphasizing that advisors may not only influence advisees but also be influenced by them in return. This view of reverse social influence enriches our understanding of social interaction mechanisms and, to our knowledge, represents the first systematic computational modeling of such dynamics in the advice-giving literature. The experimental design is rigorous and the logic across the four progressive studies is coherent. The authors build and validate the cognitive mechanism of alignment bias with clarity.

However, the manuscript can be further improved in the following respects:

1. On computational modeling:

The authors have not conducted parameter recovery prior to model fitting. This is strongly recommended to ensure that the model can reliably recover the underlying parameters, especially given the observed correlation between β and ω parameters, which raises concerns about potential parameter redundancy. Additionally, model recovery should be conducted to assess whether the models are distinguishable based on key behavioral patterns. Moreover, while the authors use hierarchical Bayesian modeling to estimate parameters of the RL models, they do not report posterior distributions, credible intervals, or MCMC diagnostics (e.g., R-hat or Gelman–Rubin statistics). Providing parameter distribution plots and convergence diagnostics would enhance the transparency and reliability of the model estimation.

2. Theoretical interpretation of acceptance-seeking:

The authors interpret alignment bias as being driven by an acceptance-seeking motivation. However, in everyday situations (e.g., choosing a restaurant or making a consumer decision) which is mentioned in the manuscript as something to be simulated, where advisors and advisees share similar expertise, accuracy may not be critical. The manuscript should further elaborate on the psychological meaning of acceptance-seeking: why are individuals motivated to seek acceptance at the cost of advice accuracy? What is the underlying psychological or social reward structure that justifies this behavior?

3. Insufficient exclusion of alternative explanations:

The current account focuses on acceptance-seeking but does not rule out other plausible motives such as rejection aversion. It is also possible that advisors treat advisees’ initial opinions as informational cues under uncertainty rather than being socially influenced. Future studies could include control conditions where the source of the initial opinion is non-social (e.g., AI agents or historical data) to distinguish social motives from cue integration strategies.

4. Missing experimental details:

Some details regarding the experimental procedures—such as the exact number of trials per condition—are missing. This hinders reproducibility and should be clearly reported.

Overall, the manuscript addresses an important and underexplored question with strong methodological foundations. With further clarification on the computational modeling and refinement of the theoretical framing, the study will make a valuable contribution to both social psychology and computational modeling literature.

Reviewer #4: ear Editor,

Summary of the paper

The authors investigate the behavioural and computational mechanisms underlying advice-alignment bias—the tendency for advisers to tailor their advice to match the recipient’s existing beliefs, thereby maximising the likelihood that the advice will be accepted. Across four well-designed experiments, they show a robust inclination among human advisers to align their recommendations with the preferences of information recipients. Their results suggest that this bias is driven primarily by deliberate conformity rather than by simple opinion contagion or anchoring effects. Reinforcement-learning (RL)–based computational modelling further confirms that advisers strategically adapt their behaviour to increase advice acceptance.

Evaluation

The study is carefully executed, and its findings will interest scholars across disciplines, from behavioural science to evolutionary ecology—for instance, in understanding the evolution of teaching behaviour, which is rare in the animal kingdom.

Suggestions for improvement

Functional implications

As someone interested in the evolutionary game dynamics of human social-learning strategies, I would welcome a discussion of the game-theoretic consequences of advice-alignment bias. If advisers merely echo advisees’ opinions, what adaptive value remains for the advisees in seeking advice at all? Elaborating on the functional rationale and downstream consequences of alignment would enrich the manuscript.

Model-recovery analysis

The RL modelling is compelling, and it is reassuring that the calibrated model reproduces human behaviour. However, the manuscript would be stronger with a full model-recovery test across the candidate models. This would clarify whether the best-fitting model is uniquely capable of explaining the data or whether alternative models can generate similar behavioural patterns.

I hope these comments are helpful.

All the best,

**Have the authors made all data and (if applicable) computational code underlying the findings in their manuscript fully available?**

Reviewer #1: Yes

Reviewer #2: Yes

Reviewer #3: None

Reviewer #4: Yes

PLOS authors have the option to publish the peer review history of their article (what does this mean? ). If published, this will include your full peer review and any attached files.

**Do you want your identity to be public for this peer review?** For information about this choice, including consent withdrawal, please see our Privacy Policy .

Reviewer #1: No

Reviewer #2: **Yes: ** Hernán Anlló

Reviewer #3: No

Reviewer #4: No

**Figure resubmission:**
---

## [Decision Letter · Decision Letter 1]

31 Oct 2025

PCOMPBIOL-D-25-00735R1

Do we advise as one likes? The alignment bias in social advice giving

PLOS Computational Biology

Dear Dr. Pan,

Thank you for submitting your manuscript to PLOS Computational Biology. After careful consideration, we feel that it has merit but does not fully meet PLOS Computational Biology's publication criteria as it currently stands. Therefore, we invite you to submit a revised version of the manuscript that addresses the points raised during the review process.

Please submit your revised manuscript within 30 days Dec 31 2025 11:59PM. If you will need more time than this to complete your revisions, please reply to this message or contact the journal office at ploscompbiol@plos.org. Please include the following items when submitting your revised manuscript:

We look forward to receiving your revised manuscript.

Kind regards,

Zhiyi Chen

Academic Editor

PLOS Computational Biology

Tobias Bollenbach

Section Editor

PLOS Computational Biology

**Additional Editor Comments:**

Thank you for the patience in the peer review process. Please address those minor concerns that the Reviewer #1 remained. This revision would not be sent for re-review to prompt the editorial process.

**Reviewers' comments:**

Reviewer's Responses to Questions

**Comments to the Authors:**

Reviewer #1: The authors took reviewer concerns seriously and made numerous revisions. I very much appreciated the added context for the social aspects of the manipulation and methodological details provided throughout.

Some points of clarification:

Title: "Do we advise as we like? The alignment bias in social advice giving" -- the mix of plural 'we' and singular 'one' in the original is confusing, because it implies that 'one' is somehow not part of the 'we'.

Line 134: "and who would provide a subjective evaluation of advice after interaction"

-- should be "their advice" or even something like "their performance in the task" as the roles and experimental structure are only made clear later

Line 219: "an attempt to avoid evaluative penalty for misaligned that proved to be incorrect." missing word -- probably this is "misaligned advice"

Line 549: "Those who indicated that they perceived the advisees as fictious would also be excluded." -- Verb tense introduces doubt / seems like speculation. Were any participants excluded due to this criteria?

Line 1197, 1247, 1264, perhaps elsewhere: Given the size of this paragraph, it may be reasonable to make a new paragraph here to aid the reader.

Line 1209: Missing '.'

Reviewer #2: I thank the authors for the careful review work they conducted, I have nothing to add. I believe the manuscript improved its quality and I'm happy with its current state.

Reviewer #3: The authors have solved all my previous questions and the quality of the manuscript has been greatly improved. I have no further concerns.

Reviewer #4: Dear Editor,

So sorry for the late review. The authors have responded all the comments I made, as well as posted by other reviewers thoroughly, and their responses were compelling. I appreciate the authors' effort in conducting new simulations and analysis for the model recovery and paramete recovery.

I do not have any further concerns nor questions. I think the manuscript is good to go as is.

I hope these comments are helpful.

All the best,

**Have the authors made all data and (if applicable) computational code underlying the findings in their manuscript fully available?**

Reviewer #1: Yes

Reviewer #2: Yes

Reviewer #3: None

Reviewer #4: Yes

PLOS authors have the option to publish the peer review history of their article (what does this mean? ). If published, this will include your full peer review and any attached files.

**Do you want your identity to be public for this peer review?** For information about this choice, including consent withdrawal, please see our Privacy Policy .

Reviewer #1: No

Reviewer #2: **Yes: ** Hernan Anllo

Reviewer #3: No

Reviewer #4: No

**Figure resubmission:**
---

## [Editor Report · Decision Letter 2]

11 Nov 2025

Dear Dr. Pan,

We are pleased to inform you that your manuscript 'Do we advise as one likes? The alignment bias in social advice giving' has been provisionally accepted for publication in PLOS Computational Biology.

Best regards,

Zhiyi Chen

Academic Editor

PLOS Computational Biology

Tobias Bollenbach

Section Editor

PLOS Computational Biology

Authors have addressed all the remaining minor concerns well.

---

## [Editor Report · Acceptance letter]

PCOMPBIOL-D-25-00735R2

Do we advise as one likes? The alignment bias in social advice giving

Dear Dr Pan,

I am pleased to inform you that your manuscript has been formally accepted for publication in PLOS Computational Biology. Your manuscript is now with our production department and you will be notified of the publication date in due course.

With kind regards,

Judit Kozma
